# Interpretability for Time Series Transformers using A Concept Bottleneck Framework

## Abstract

There has been a recent push of research on Transformer-based models for long-term time series forecasting, but the interpretability of these models remains largely unexplored. To address this gap, we develop a framework based on Concept Bottleneck Models. We modify the training objective to encourage a model to develop representations similar to predefined interpretable concepts using Centered Kernel Alignment. We apply the framework to the Vanilla Transformer and Autoformer, and present an in-depth analysis on synthetic data and on a variety of benchmark datasets. We find that the model performance remains mostly unaffected, while the model shows much improved interpretability. Additionally, interpretable concepts become local, which makes the trained model easily intervenable. We demonstrate this with an intervention after applying a time shift to the data.

## 1 Introduction

Transformers show great success for various types of sequential data, including language (Devlin, 2018; Brown, 2020), images (Dosovitskiy et al., 2021; Liu et al., 2021), and speech (Baevski et al., 2020). Their ability to capture long-term dependencies has triggered substantial interest in applying them to time-series, which are naturally sequential, and in particular to the challenging task of long-term time series forecasting. Transformer-based architectures, indeed, often show superior performance on this task (Zhou et al., 2021; 2022; Wu et al., 2021; Ni et al., 2023; Chen et al., 2024), for an overview we refer to Wen et al. (2023).

However, due to their deep and complex architecture, Transformers are difficult to interpret, which is especially important in high-stakes domains such as finance and energy demand prediction. There is a large body of work in the field of explainable AI to interpret neural networks (Bereska & Gavves, 2024), or increase their interpretability, including the approach of Concept Bottleneck Models (CBMs; Koh et al., 2020). This approach relies on the idea of constraining the model such that it first predicts human-interpretable concepts, and then uses only these concepts to make the final prediction. CBMs and their variants have become popular in various fields, especially in computer vision, but are so far unexplored in the context of time series forecasting.

In this paper, we propose a training framework to make any time series Transformer into a Concept Bottleneck Model using time-series specific, yet domain-agnostic concepts, as shown in Figure 1. A key aspect of our training framework is to leave the model's architecture intact, while encouraging the learned representations to be similar - but not identical - to the interpretable concepts. We measure similarity with Centered Kernel Alignment (CKA; Kornblith et al., 2019) and include it in the loss function. The first concept is a simple, linear surrogate model and the second is time information (e.g. hour-of-day). Note that we propose a *global* interpretability method, which improves identifying and localizing high-level concepts in the model's internal mechanisms, and is not comparable to local post-hoc interpretability methods such as SHAP, LIME, or attention-based visualizations which explain individual predictions.

We apply our concept bottleneck framework to three types of models: the vanilla Transformer (Vaswani et al., 2017), the Autoformer (Wu et al., 2021) and the FEDformer (Zhou et al., 2022). Across extensive experiments on seven datasets, we show that our setup results in models that are more interpretable while the overall performance remains largely unaffected – in many cases surpassing results from the original

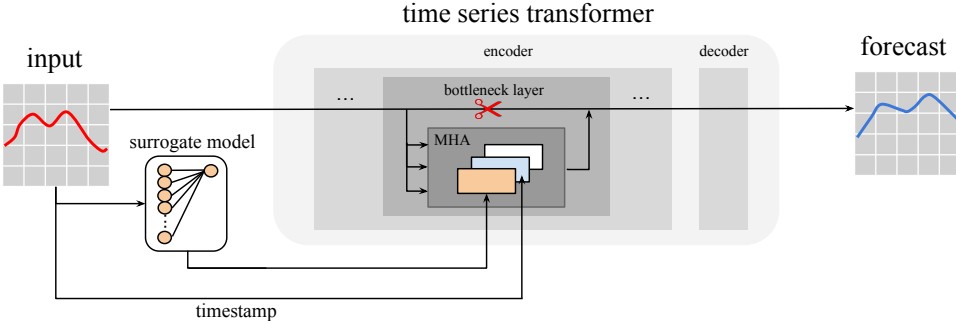

Figure 1: Overview of the concept bottleneck framework. We use one encoder layer as bottleneck, and train its similarity with pre-defined, interpretable concepts. In this example, the bottleneck is located in the multi-headed attention (MHA) block, of which one head is trained to be similar to the surrogate model, another head to the timestamps, and the final head remains untouched.

Autoformer paper. Furthermore, we explicitly test the faithfulness of the obtained interpretations with an intervention study.

Our contributions are summarized as follows:

1. We propose a novel training framework to increase the interpretability of Transformers for time series.

2. We demonstrate the feasibility of applying this framework to time-series Transformers by conducting extensive experiments on three types of Transformers and seven datasets, and identify interpretable concepts in each of these Transformers.

3. We assess the faithfulness of the interpretability analysis by performing an activation patching experiment, and obtain evidence that the identified components (in the concept bottleneck) indeed have the hypothesized unique and causal role in the predictions of the target model.

## 2 Background and Related Work

This paper combines and builds upon foundational works from different fields, including CBMs, knowledge transfer with CKA and time series Transformers. CBMs have been applied to time series before (Ferfoglia et al., 2024), but not with the same interpretable concepts. Likewise, the similarity index CKA has been used before to transfer knowledge between models (Tian et al., 2023), yet, to the best of our knowledge, it has not been used to construct a CBM. This makes our work a unique contribution at the intersection of (mechanistic) interpretability, concept learning, and time series forecasting.

### 2.1 Concept Bottleneck Models

Concept Bottleneck Models (CBMs; Koh et al., 2020) have emerged as promising interpretable models (Poeta et al., 2023). The concept bottlenecks constrain the model to first predict interpretable concepts, and then use only these concepts in the final downstream task. They are shown to be useful in multiple applications, such as model debugging and human intervention. The bottleneck allows for explaining which information the model is using and when it makes an error due to incorrect concept predictions.

One of the shortcomings of standard CBMs is that concept annotations are needed during training to learn the bottleneck. To alleviate this issue, variants have been proposed to learn the concept bottleneck itself too using other datasets, large language models or other, multimodal models (Yuksekgonul et al., 2023; Oikarinen et al., 2023; Yang et al., 2023). Shang et al. propose incremental concept discovery, such that

missing concepts to any concept bank can be identified. However, concept labels do not necessarily contain all information needed to accurately perform the downstream task, and can therefore decrease the task accuracy (Mahinpei et al., 2021). Therefore, Zarlenga et al. (2022) propose Concept Embedding Models, where concepts are represented as vectors, such that richer and more meaningful concept semantics can be captured.

CBMs can suffer from information leakage, where the model makes use of additional information in the concept space rather than the concept information itself (Mahinpei et al., 2021; Havasi et al., 2022). This can occur when the model is jointly trained on the concept prediction and down-stream task, and if it uses soft concepts (i.e. numerical representations with values between 0 and 1). Information leakage compromises the interpretability and intervenability in soft CBMs, but it can be addressed when introducing a side-channel (Havasi et al., 2022) or by alignment of the model's representation with an underlying data generation process using disentangled representation learning (Marconato et al., 2022).

CBMs and their variants are usually applied to the field of computer vision, and less frequently to natural language (Tan et al., 2024), graphs (Barbiero et al., 2023) or tabular data (Zarlenga et al., 2022). In principle, the methodology can be applied to time series as well, but defining high-level, meaningful concepts is challenging. Ferfoglia et al. (2024) use Signal Temporal Logic (STL) formulas as concept embeddings for time series to convert them into natural language, and use these concepts as bottleneck for anomaly detection.

## 2.2 Knowledge Transfer with Centered Kernel Alignment

Inspired by neuroscience, CKA measures the similarity between different representations from neural networks (Kornblith et al., 2019). CKA captures intuitive notions of similarity between representations. To obtain the score, firstly, the similarity between every pair of examples in each representation separately is measured using a pre-defined kernel, and then the obtained similarity structures are compared. We refer to Kornblith et al. (2019) for more details.

The CKA score can be used to transfer knowledge between different models when included in the loss function Tian et al. (2023). In this work, the authors study knowledge distillation between a teacher and student model, and incorporate CKA into the loss function to transfer feature representation knowledge from the pretrained model (teacher) to the incremental learning model (student). The goal is to encourage the incremental learning model not to forget previously learned knowledge, while continuously learning new knowledge, so that it is able to adapt to a dynamic environment (Parisi et al., 2019).

Note that we are well aware that other representational similarity (matrix-based) measures exist, as for instance described by Klabunde et al. (2025). Nevertheless, we believe CKA is a sensible choice to use in our work, as it has been shown to be effective in knowledge transfer and is also popular and well-studied by the deep learning community (Williams, 2024).

## 2.3 Time Series Transformers

Time series transformers for long-term time series forecasting, such as the Autoformer and FEDformer, obtain two types of input: (1) *data values* $\boldsymbol{X} \in \mathbb{R}^{I \times d}$, and (2) *timestamps* $\boldsymbol{T} \in \mathbb{R}^{I \times 4}$. More specifically, they can be regarded as a function $f : \mathbb{R}^{I \times d} \times \mathbb{R}^{I \times 4} \times \mathbb{R}^{O \times 4} \to \mathbb{R}^{O \times d}$, where $I$ is the number of input time steps, $O$ is the number of future time steps, and $d$ is the number of variables in the time series. The additional four dimensions of timestamps $\boldsymbol{T}$ represent four time features, namely *hour-of-day, day-of-week, day-of-month*, and *day-of-year*. The future timestamps are also provided, for which the model should forecast the future data values. Note that we explicitly introduce a notation for the timestamps to later define the CKA scores and the intervention.

The Autoformer is a Transformer-based model for long-term time series forecasting, as introduced by Wu et al. (2021). It is a common and influential work among the time series Transformers, and serves as a good representative for our framework. The model's encoder-decoder architecture is inspired from time series decomposition, and contains two major modifications to the original Transformer architecture. Firstly, the Autoformer contains decomposition blocks, such that the long-term trend information can be separated

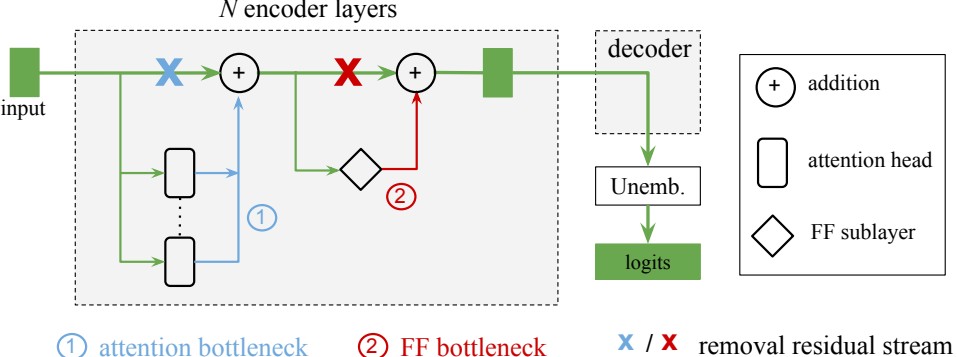

Figure 2: Architecture of Autoformer with a concept bottleneck in the attention mechanism (blue) or the FF network (red). Note that the residual connection is removed at the location of the bottleneck (and the 'residual stream' thus interrupted). Visualisation inspired by Rai et al., 2024.

from the seasonal information. Secondly, Autoformer employs an auto-correlation mechanism instead of self-attention, such that similarities can be measured on subseries [1]. An overview of the architecture from the original paper is provided in Appendix A.

## 3 Method

We propose a training framework to make any Transformer model interpretable by including a bottleneck based on knowledge transfer with CKA (Kornblith et al., 2019), as shown in Figure 2. The main idea is that we assign one of the encoder layers to be the *concept bottleneck*; representations in the bottleneck are subject to a soft constraint of being as similar as possible to predefined interpretable concepts. To this end, we calculate CKA scores with the interpretable concepts, and include these scores in the loss function.

### 3.1 Loss Function

The loss function should encourage the model to represent the interpretable concepts in the bottleneck layer. Therefore, we add a term $\mathcal{L}_{CKA}$ based on the CKA scores of the bottleneck and the interpretable concepts (Eq. 2). In particular, low similarity between the bottleneck and the interpretable concepts results in a higher value for $\mathcal{L}_{CKA}$. The total loss function $\mathcal{L}_{Total}$ (Eq. 1), then, is a weighted average of the Mean Squared Error (MSE) loss $\mathcal{L}_{MSE}$ and the CKA loss $\mathcal{L}_{CKA}$:

$$\mathcal{L}_{Total} = (1 - \alpha)\ \mathcal{L}_{MSE} + \alpha\ \mathcal{L}_{CKA}, \tag{1}$$

$$\mathcal{L}_{CKA} = 1 - \frac{1}{c} \sum_{i=1}^{c} CKA_i, \tag{2}$$

where $\alpha$ is a hyperparameter, $c$ is the number of concepts, and $CKA_i \in [0, 1]$ is the CKA score (see Section 3.2) between the model's representation and concept $i$.

### 3.2 Interpretable Concepts in the Bottleneck

In this section, we describe how to calculate the CKA score to measure the presence of a concept.

**Location bottleneck.** We assign one encoder layer to be the bottleneck layer, because the encoder focuses on modelling seasonal information. Within the bottleneck layer, the latent representations can be taken from two different types of blocks: the auto-correlation block ($\tau = Att$) and the feed-forward block ($\tau =$

---

[1]Note that the *auto-correlation block* is an implementation of the *attention block*, so we use the two terms interchangeably.

$FF$). These two options are illustrated in Figure 2. We assign $c$ interpretable concepts over the latent representations, with the goal of teaching the corresponding model component to represent the pre-defined interpretable concepts.

Since the attention block is multi-headed, different heads naturally form the components of the attention bottleneck. Moreover, the components need to be divided between the heads, which would be convenient when the number of heads is a multiple of the total number of concepts to maintain a uniform concept per head ratio. For the feed-forward bottleneck, we define the components to be slices from its output, such that stacking the components results in the original output.

**Interpretable concepts.** We use two domain-agnostic interpretable concepts which can be used for forecasting, namely: (1) a simple, human-interpretable surrogate forecasting model, (2) the input timestamps recorded with the time series.

1. We use a simple autoregressive model (AR) as a surrogate model, which predicts the next future value as a linear combination of its past values. This model is transparent, and the attribution of each input feature to the output can be simply interpreted by its weight. This concept can also be regarded as a baseline for the forecasting performance. The model is fit to the same training data as the Autoformer model.

2. We use the hour-of-day feature from the timestamps $\boldsymbol{T}$ as interpretable time concept, denoted by $\boldsymbol{T}_{hourofday}$. This provides the bottleneck with a simplified notion of time.

**Removal of residual connection.** Any Autoformer layer contains residual connections around the auto-correlation and feed-forward blocks. To ensure that all information passes through the bottleneck, we remove the residual connection around the bottleneck, potentially at the cost of a loss in performance. Otherwise, any concept, including the interpretable concepts, can be passed through the residual connection and compromise the bottleneck.

In the scenario that the number of components is equal to the number of interpretable concepts ($c = 2$), the construction of the bottleneck limits learning domain-specific features from the data, other than the interpretable concepts. Therefore, we perform experiments where we allow an extra component in the bottleneck to not learn any pre-defined concept ($c = 3$). In other words, the extra component serves as a *side-channel* or *free component*. No CKA loss is calculated using this component, and therefore this training set-up can be regarded as semi-supervised. The free component may partly restore what we lost by removing the residual connection, but with the advantage that we can monitor which information goes through it, and even visualize it (as in Section 4.3.2).

We refer to Appendix C for a more detailed description of the concept bottleneck framework and to Appendix E for detailed visualizations of both types of bottlenecks.

### 3.3 Implementation details.

In our experiments, we use an Autoformer with three encoder layers, of which the bottleneck layer is at position $\ell = 2$. Similar to the original Autoformer paper, we use one decoder layer, employ the Adam optimizer (Kingma & Ba, 2017) with an initial learning rate of $10^{-4}$, and use a batch size of 32. The training process is early stopped within 25 epochs. All experiments are repeated five times on different seeds, using hyperparameter $\alpha = 0.3$.

## 4 Experiments

We evaluate our framework on two models and seven datasets, including synthetic and real-world data. The six real-world benchmarks consider the domains of energy, traffic, economics, weather, and disease, similar to Wu et al. (2021). These datasets are multivariate, and the task is to predict the future values of all variates. For example, the electricity dataset consists of hourly measurements of the electricity consumption of 321 customers from 2012 to 2014. For more information on the datasets, we refer the reader to Appendix B. We

apply the experiments to the vanilla Transformer and Autoformer. First, we train a simple AR model on the same data, so that its outputs can be used to align the representations of the bottleneck. Then, we train the Transformers with and without bottleneck, using different configurations for the bottleneck.

## 4.1 Synthetic Data

To show the general applicability of the bottleneck framework, we first train an Autoformer on a synthetic time series. In particular, we generate the dataset as the sum of different sines using the function $f_{Total}$ with time $t$ as follows:

$$f_{Total}(t) = f_1(t) + f_2(t) + f_3(t),$$

where:

$$f_1(t) = \sin(2\pi t),$$
$$f_2(t) = \frac{1}{2}\sin(4\pi t + \frac{\pi}{4}),$$
$$f_3(t) = \frac{1}{4}\sin(6\pi t + \frac{\pi}{2}) + \epsilon_t.$$

Note that all functions $f_1, f_2$ and $f_3$ follow a periodic structure, and $f_3$ contains random noise $\epsilon$ from a normal distribution with standard deviation of 0.2.

Each concept in the bottleneck is defined as one of the underlying functions (i.e., $f_1$, $f_2$ or $f_3$), for which the ground-truth is known by construction. For hyperparameter $\alpha = 0.8$ (see Section 3.1), we find that the model is able to forecast well, while achieving very high similarity scores. That is, the model obtains a Mean Squared Error (MSE) of $0.36 \pm 0.17$ and Mean Absolute Error (MAE) of $0.46 \pm 0.12$ on 5 different seeds. See Figure 3 for a sample forecast on the test data and the CKA scores of the model's representations with the concept representations. The heads in the bottleneck `layer1` show high similarity for their respective concepts, e.g. a score of 0.93 for the head trained on $f_1$ (recall that CKA scores range from 0 for totally dissimilar to 1.0 for identical, although potentially scaled and rotated). We refer to Appendix L for more results on the synthetic dataset.

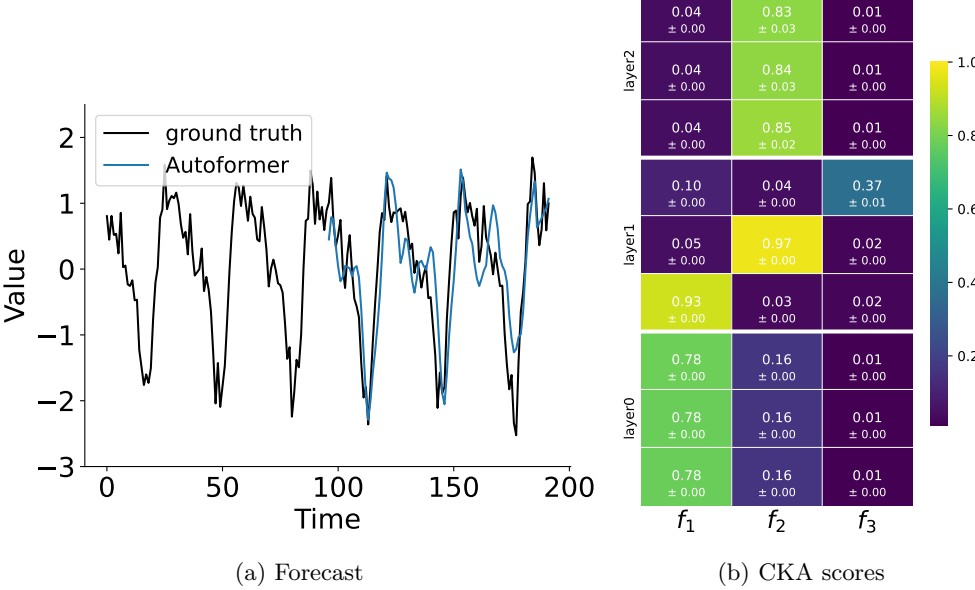

(a) Forecast

(b) CKA scores

Figure 3: Forecast and CKA scores of the attention bottleneck Autoformer on synthetic data. The scores are calculated on three batches of size 32 from the electricity test data.

### 4.2 Performance Analysis Benchmarks

Table 1 shows the performance of the Autoformer with our bottleneck on the benchmark datasets, compared to the AR surrogate model (i.e. the first interpretable concept) and Wu et al. (2021) (i.e. the original Autoformer model). Note that the bottleneck models are trained with a free component, i.e., $c = 3$, and the original Autoformer is of a different size (two encoder layers with eight heads per layer). Visualizations of the forecasts from these models are shown in Appendix F.

Table 1: Performance of different models. For both metrics, it holds that a lower score indicates a better performance, where the best results are **bold**, and the second-best are underlined.

| | Att bottleneck | | FF bottleneck | | No bottleneck | | AR | | Wu et al. | |
|---|---|---|---|---|---|---|---|---|---|---|
| | MSE | MAE | MSE | MAE | MSE | MAE | MSE | MAE | MSE | MAE |
| Electricity | 0.231 | 0.338 | 0.207 | 0.320 | 0.280 | 0.368 | 0.497 | 0.522 | **0.201** | **0.317** |
| Traffic | 0.642 | 0.393 | **0.393** | **0.377** | 0.619 | 0.387 | 0.420 | 0.494 | 0.613 | 0.388 |
| Weather | 0.290 | 0.354 | 0.271 | 0.341 | 0.269 | 0.344 | **0.006** | **0.062** | 0.266 | 0.336 |
| Illness | 3.586 | 1.313 | 3.661 | 1.322 | 3.405 | 1.295 | **1.027** | **0.820** | 3.483 | 1.287 |
| Exchange rate | 0.195 | 0.323 | 0.155 | 0.290 | 0.152 | 0.283 | **0.082** | **0.230** | 0.197 | 0.323 |
| ETT | 0.177 | 0.282 | 0.174 | 0.280 | 0.155 | 0.265 | **0.034** | **0.117** | 0.255 | 0.339 |

We find that including a bottleneck (either **Att bottleneck** or **FF bottleneck**) outperforms **Wu et al.** for three datasets (traffic, exchange rate and ETT), and stays within 5% of the MSE and MAE for the other three datasets. Surprisingly, the surrogate AR model outperforms the other models for most datasets w.r.t. both MSE and MAE, even though this model is very simple.[2] More detailed results are presented in Appendix G and H, where the first includes the results for bottlenecks without free component (including the standard deviation for different seeds), and the latter includes a sensitivity analysis to hyperparameter $\alpha$.

Similar to the Autoformer, the vanilla Transformer and FEDformer with a bottleneck outperform models without bottleneck for some datasets, see Appendix J and K, respectively.

### 4.3 Interpretability Analysis Benchmarks

To demonstrate the impact of the bottleneck on model interpretability, we first conduct a CKA analysis on the bottleneck layer with the corresponding interpretable concepts, and then visually demonstrate how each component contributes to the final forecast.

#### 4.3.1 CKA Analysis

To test the extent to which the bottleneck represents the interpretable concepts, we calculate the CKA scores of the model's representations with the concept representations. The scores of the feed-forward bottleneck on the electricity dataset are shown in Figure 4 (see Appendix H for more scores on the Autoformer). Note that the bottom, middle and upper layer of `layer1` correspond to the AR, hour-of-day, and free component of the bottleneck, respectively.

The scores show that the representations in the bottleneck layer are much more similar to the intended concepts than the representations from the model without bottleneck: there is a score of 0.94 for the AR model, and 1.00 for the hour-of-day feature, whereas the model without bottleneck does not show high similarity to the interpretable concepts. This indicates that the training framework can encourage the components to form representations that are perfectly similar to the interpretable concepts. Additionally, note that the CKA scores of other layers than the bottleneck layer are also higher in Fig. 4b, which indicates

---

[2]Note that the phenomenon that simple models sometimes beat time series Transformers (Zeng et al., 2022) has been observed before. There has been a vivid discussion about the relevance of these results, for instance here. These discussions are beyond the scope of our paper, which rather targets interpretability of time series Transformers. For more information on the effect of AR as surrogate model, see Appendix M.

that these other model components also learn to represent the interpretable concepts. This does not affect the interpretability of the bottleneck layer itself (we refer to Section 4.4).

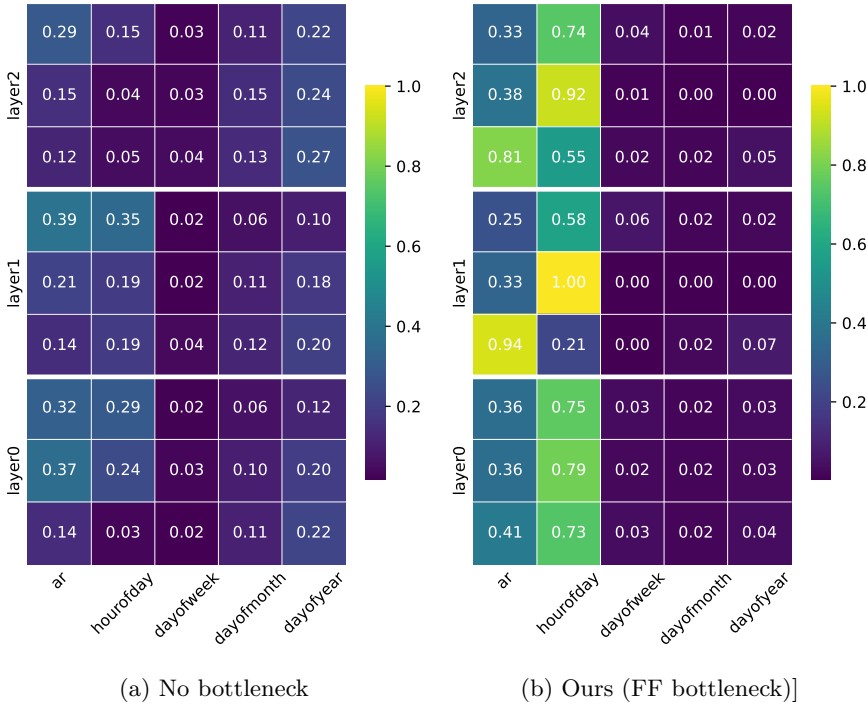

(a) No bottleneck
(b) Ours (FF bottleneck)]

Figure 4: CKA scores on different concepts for the encoder of the vanilla Transformer without bottleneck and with FF bottleneck. Both models contain three heads per layer. The first component of `layer1` (lower row) of the attention bottleneck is trained to be similar to AR, and the second component (middle row) to the hour-of-day concept. The scores are calculated on three batches of size 32 from the electricity test data. Recall that CKA is defined on a scale from 0 to 1, where 1 denotes perfect similarity.

### 4.3.2 Component Visualizations

Interestingly, because the components we consider all read and write from the 'residual stream' (Elhage et al., 2021), we can visualize the contributions to the final prediction of each component separately by applying the entire decoder to the component representations (Decoder Lens method, Langedijk et al. (2023)). This way, we obtain visualizations of the contributions of each component in the bottleneck, see Figure 5. We obtain the output from the full bottleneck by applying the decoder to the output of the bottleneck (after performing layer normalization). The output from each component individually is obtained by masking the other components with zero (close to the mean).

From Figure 5a and 5b we see that the different bottleneck components are similar to the concepts they were trained on. In particular, the first component shows a forecast with correct periodicity and few irregularities, similar to the actual forecast from the AR model. Likewise, the second component shows a periodicity to the actual hour-of-day feature. The third component is not trained to be similar to an interpretable concept, and seems to pick up on the high-frequency patterns in the data, e.g., the low, second peak in the forecast. This observation is further strengthened by Figure 5f, which shows that the final forecast consists of many high-frequency patterns when using only the third component from the bottleneck. We find similar component visualizations on the vanilla Transformer, see Appendix J.3.

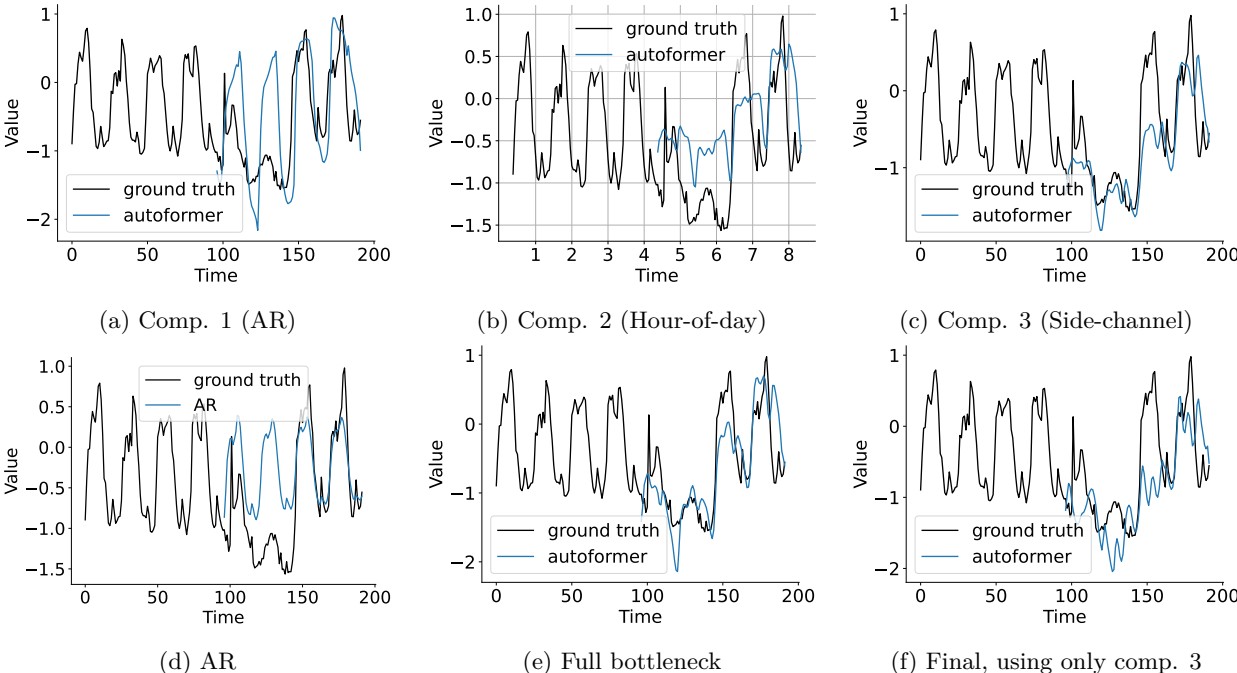

Figure 5: Forecasts from individual bottleneck components by masking the other components with zero in 5a, 5b and 5c (FF bottleneck on electricity data). The first half of the ground truth forms the input to the model. Note that the horizontal axes are the same across all figures, but Figure 5b contains a grid of days instead of numbered hours. Figure 5d shows the forecast made by the surrogate model AR; Figure 5e shows the forecast of the entire layer (i.e., all components together), and 5f shows the forecast of the final layer when only the third component is used in the bottleneck layer. Note the difference between Figures 5c and 5f, where we decode from the bottleneck and the final layer, respectively.

## 4.4 Intervention

The main benefit of interpreting trained models is gaining a deeper understanding and, possibly, more control of the model's behavior. This can be useful in the scenario of out-of-distribution data at inference time. If the data changes in features that can be interpreted in the model, it is feasible to intervene locally in these concepts to exclusively employ the model with data from its training distribution. Additionally, an intervention can be regarded as a causal interpretability test, where a successful intervention indicates a successful representation of the concept of interest.

To show such benefit of our framework, we evaluate the trained model on data with shifted timestamps and compare it with performing an intervention on the shifted concept. This experiment is inspired by activation patching (or causal tracing, Meng et al., 2023), where causal effects of hidden state activations are researched by evaluating the model on clean and corrupted inputs.

More specifically, we delay the input timestamps $\boldsymbol{T} \in \mathbb{R}^{I \times 4}$ with a fixed number of hours to obtain the shifted timestamps $\widetilde{\boldsymbol{T}}$, so that the learned patterns associated to the hour-of-day feature are misleading. We run the model on both types of timestamps, and perform an intervention in the bottleneck by substituting the activations based on the shifted time with the activations based on the original time, see Figure 6 for an overview.

We perform the intervention experiment with the electricity dataset, and perform shifts of up to and including 23 hours. We compare the performance of the intervention with out-of-the-box performance of the same model on the shifted dataset. The results of the vanilla Transformer and Autoformer are shown in Figure 7.

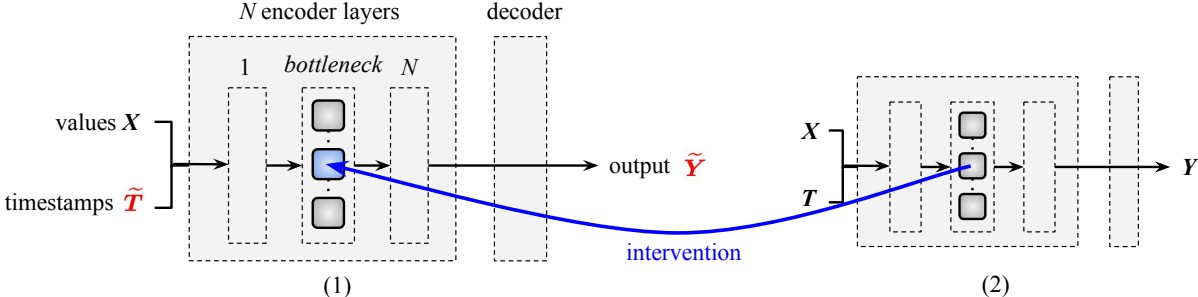

Figure 6: Overview of the intervention experiment, where we run the Autoformer on time-shifted input (1), but replace the activations of the hour-of-day component in the bottleneck with the activations from a previous run (2) on the original, unshifted timestamps $T$.

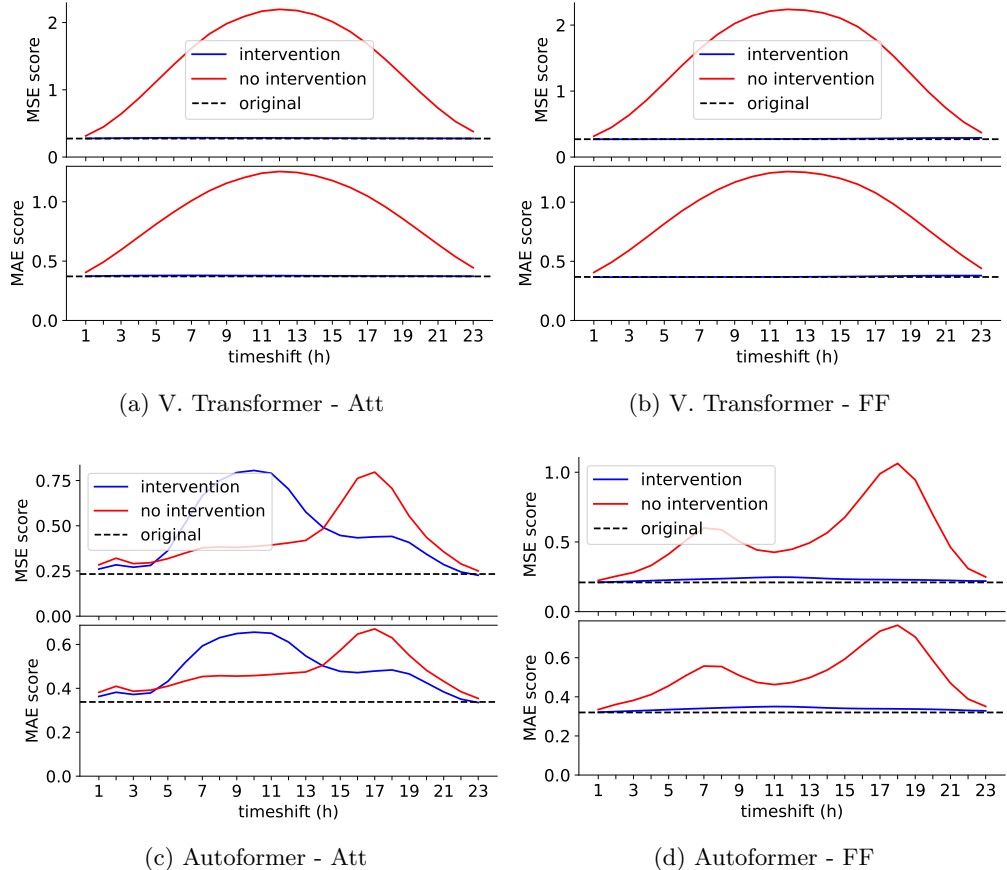

(a) V. Transformer - Att

(b) V. Transformer - FF

(c) Autoformer - Att

(d) Autoformer - FF

Figure 7: Performance of the attention bottleneck (Att) and feed-forward bottleneck (FF) in the vanilla (V.) Transformer and Autoformer on electricity data with shifted timestamps. The dashed line represents the performance of the same model on the original data, i.e., with no timeshift.

Remarkably, the intervention on the vanilla Transformer achieves the original performance for all timeshifts. The same holds for the Autoformer feed-forward bottleneck, but not for the attention bottleneck (Fig. 7c). This indicates that, overall, the bottleneck models effectively learn to represent the hour-of-day concept in the dedicated bottleneck component. Most interestingly, the models only utilise this interpretable concept in the bottleneck layer, but not in other encoder layers (because the experiment only intervenes in the bottleneck).

Note that the intervention hurts the performance for smaller timeshifts in the Autformer with attention bottleneck. This indicates that the choice of bottleneck location is relevant for the Autoformer, and suggests that the concept of time might be represented in a more complex manner in the attention heads. Presumably, intervening in an individual head within the multi-headed auto-correlation mechanism provides more unforeseen consequences than intervening in the slice of a linear layer due to the increased complexity.

## 5 Discussion and Conclusions

In this work, we propose a training framework based on Concept Bottleneck Models to enforce interpretability of time series Transformers. We introduce a new loss function based on the similarity score CKA of the model's representations and interpretable concepts. We apply our framework to the vanilla Transformer and Autoformer model using synthetic data and six benchmark datasets. Our results indicate that the overall performance remains unaffected, while the model's components become more interpretable. Additionally, it becomes possible to perform a local intervention when employing the model after a temporal data shift.

The main limitation of our concept bottleneck framework is that interpretable concepts have to be decided on before training, which might require domain knowledge. Representations for these concepts have to be available during training. However, domain-agnostic concepts such as the AR surrogate model and hour-of-day information are also sufficient. Additionally, our framework increases computational complexity. This might be problematic if the size of the architecture increases.

An interesting direction for future research would be to optimize the number and type of interpretable concepts in the bottleneck, and extend the framework to other modalities. We trained mostly using two domain-agnostic concepts (AR and hour-of-day), but including more concepts, possibly domain-specific, would be very interesting. For example, one could consider choosing speech and music concepts for audio time series. Additionally, the framework should also work for transformers in other modalities, e.g., language and vision, although these models are usually of larger size. We hope our work contributes to a deeper understanding of (time series) transformers and their behavior in different fields. In particular, recent progress in the field of mechanistic interpretability is based on the observation that the residual stream of the Transformer encourages modular solutions, which enables localized concepts or specialized circuitry to perform a specific task. Instead of relying on post-hoc localization of these concepts, our paper presents a demonstration that we can encourage locality of concepts, without a significant loss in performance.

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

# A   Autoformer Architecture

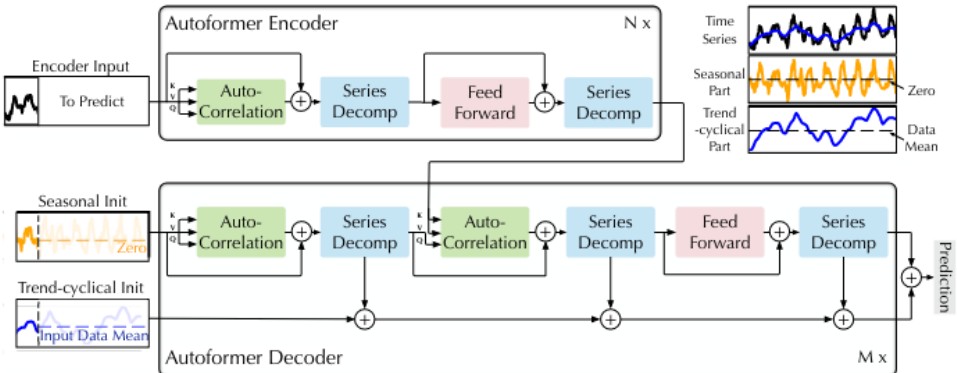

Figure 8: Overview of the Autoformer architecture from Wu et al. (2021). The Autoformer consists of an encoder and decoder, which are built from auto-correlation, series decomposition and feed-forward blocks. The encoder eliminates the long-term trend-cyclical part, while the decoder accumulates the trend part.

# B   Datasets

We evaluate the Autoformer model on six real-world benchmarks, covering the five domains of energy, traffic, economics, weather, and disease. We use the same datasets as Wu et al. (2021), and provide additional information in Table 2, as given in the original Autoformer paper.

Table 2: Descriptions of the datasets, as given by Wu et al. (2021) and shared online. 'Pred len' denotes the prediction length used in our experiments.

| Dataset | Pred len | Description |
|---|---|---|
| Electricity | 96 | Hourly electricity consumption of 321 customers from 2012 to 2014. |
| Traffic | 96 | Hourly data from California Department of Transportation, which describes the road occupancy rates measured by different sensors on San Francisco Bay area freeways. |
| Weather | 96 | Recorded every 10 minutes for 2020 whole year, which contains 21 meteorological indicators, such as air temperature, humidity, etc. |
| Illness | 24 | Includes the weekly recorded influenza-like illness (ILI) patients data from Centers for Disease Control and Prevention of the United States between 2002 and 2021, which describes the ratio of patients seen with ILI and the total number of the patients. |
| Exchange rate | 96 | Daily exchange rates of eight different countries ranging from 1990 to 2016. |
| ETT | 96 | Data collected from electricity transformers, including load and oil temperature that are recorded every 15 minutes between July 2016 and July 2018. |

# C  Formalization of Concept Bottleneck Framework

Any time series Transformer obtains two types of input: (1) *data values* $\boldsymbol{X} \in \mathbb{R}^{I \times d}$, and (2) *timestamps* $\boldsymbol{T} \in \mathbb{R}^{I \times 4}$. The transformer consists of an encoder and a decoder, which are both constructed from one or multiple layers. Any encoder layer contains two sub-layers: a multi-head attention mechanism (Att) and a fully connected neural network (FF). Every sub-layer contains a residual connection around it. More specifically, the output $\boldsymbol{X}^\ell$ of any encoder layer $\ell$ is:

$$\boldsymbol{X}^\ell = \text{Encoder}(\boldsymbol{X}^{\ell-1})$$
$$= \text{LayerNorm}(\text{FF}(\boldsymbol{S}^\ell) + \boldsymbol{S}^\ell),$$
$$\boldsymbol{S}^\ell = \text{LayerNorm}(\text{Att}(\boldsymbol{X}^{\ell-1}) + \boldsymbol{X}^{\ell-1}),$$

where

$$\text{FF}(\mathbf{x}) = \max(0, \mathbf{x}\boldsymbol{W}_1 + \mathbf{b}_1)\boldsymbol{W}_2 + \mathbf{b}_2,$$
$$\text{Att}(\mathbf{x}) = \boldsymbol{W}_0 \cdot \text{Concat}\left(\text{h}_1(\mathbf{x}), \ldots, \text{h}_h(\mathbf{x})\right).$$

For future reference, we denote the output of the feed-forward module as follows: $\text{FF}(\boldsymbol{S}^\ell) = \boldsymbol{Z}^\ell \in \mathbb{R}^{d_1 \times d_2}$. We omit the definition of the decoder, because our bottleneck framework does not include it. Note that the exact implementation of each (sub-)layer depends on the type of Transformer.

## C.1  Bottleneck Layer

We assign one encoder layer to be the bottleneck and construct it such that it contains $c$ latent representations or *components*, i.e., $(\boldsymbol{H}_i)_{i=1}^c$. Depending on the bottleneck type $\tau$, these latent representations are either taken from the attention mechanism or the feed-forward module. More specifically:

$$\boldsymbol{H}_i = \begin{cases} \text{h}_i(\mathbf{x}) & \text{if bottleneck type } \tau = \text{Att,} \\ \boldsymbol{Z}_i & \text{if bottleneck type } \tau = \text{FF.} \end{cases}$$

Since the attention block is multi-headed, different heads naturally form the components of the attention bottleneck. For the feed-forward bottleneck, we define the components to be slices (in $d_1$) from its output $\boldsymbol{Z}$, such that stacking the components results in the original output.

Note that the residual connection around the corresponding bottleneck component is removed, and that each component $\boldsymbol{H}_i$ should represent a pre-defined interpretable concept.

## C.2  Intervention

In the intervention experiment, we shift the time stamps $\boldsymbol{T}$ to obtain $\widetilde{\boldsymbol{T}}$. The key aspect of the experiment is to run the Transformer on the shifted time stamps $\widetilde{\boldsymbol{T}}$, and replace the input representations $\widetilde{\boldsymbol{X}}^{b-1}$ of the bottleneck layer $b$ with $\boldsymbol{X}^{b-1}$ (based on $\boldsymbol{T}$), but only in the component that represents the time concept.

More specifically, if type $\tau = \text{Att}$, we intervene on the attention block in the bottleneck as follows:

$$\text{Att}(\mathbf{x}, \widetilde{\mathbf{x}}) = \boldsymbol{W}_0 \cdot \text{Concat}\left(\text{h}_1(\widetilde{\mathbf{x}}), \text{h}_2(\mathbf{x}), \text{h}_3(\widetilde{\mathbf{x}})\right),$$

and, if type $\tau = \text{FF}$, as follows:

$$\text{FF}(\mathbf{x}, \widetilde{\mathbf{x}}) = \text{Stack}(\widetilde{\boldsymbol{Z}}_1, \boldsymbol{Z}_2, \widetilde{\boldsymbol{Z}}_3).$$

In both functions we make use of the fact that the time concept is represented in the second component, and there are three components in total. This intervention can be done in the bottleneck only, because, by construction, its location of the concept representations is known.

# D Detailed Overview Bottleneck Architecture

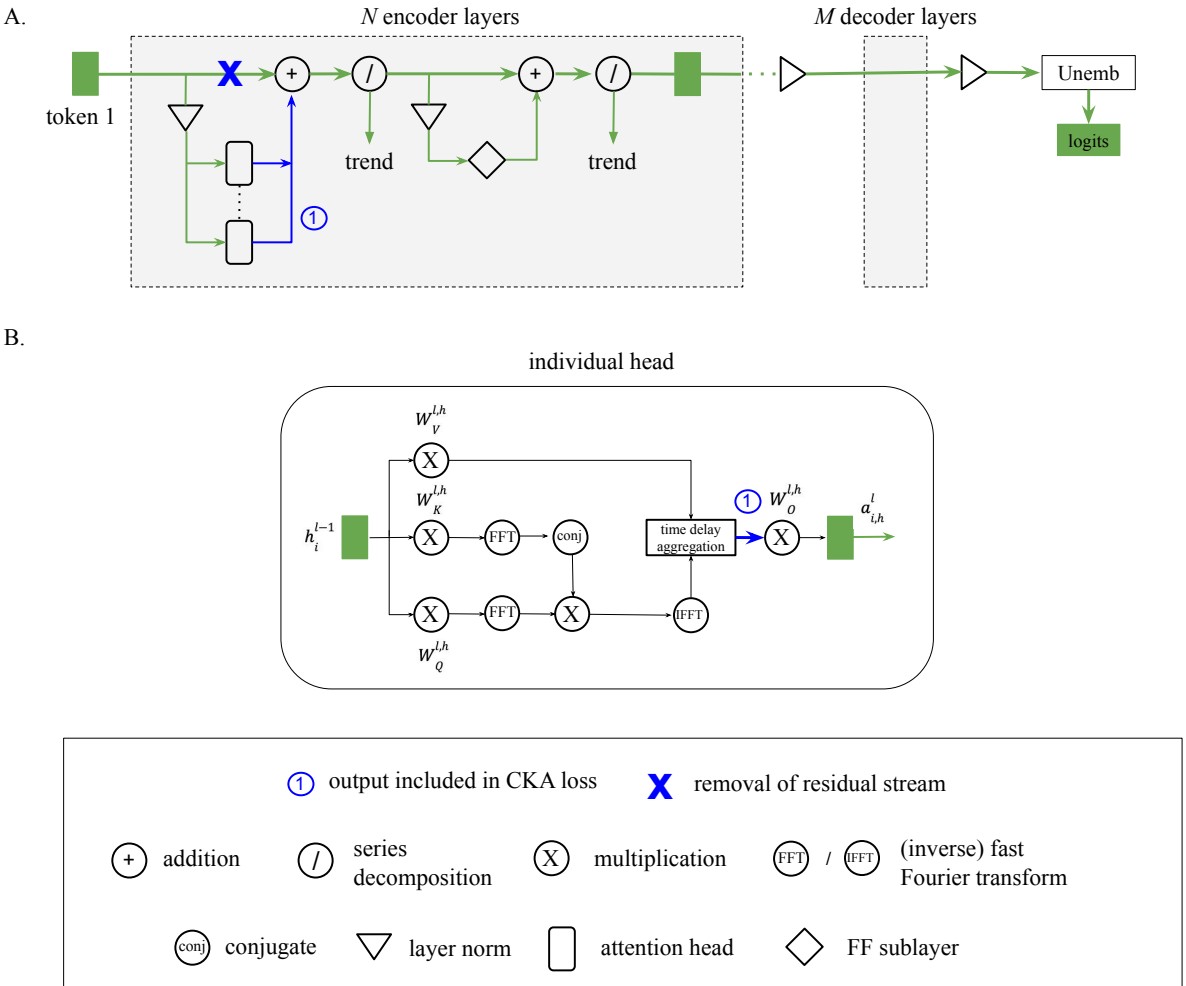

Figure 9: Architecture of Autoformer with a concept bottleneck in the attention mechanism. Figure A shows the total architecture, including the removal of the residual stream at the location of the attention block. Figure B shows the overview of an individual attention head $i$ with its input representation $h_i^{l-1}$ and output activation $a_{i,h}^l$. The representations after the time delay aggregation block are used in the bottleneck, because these correspond to the output of head $i$. Note that in practice, the individual head representations do not exist anymore after multiplication with the weights $W_o^{l,h}$, as this matrix projects the activations of all heads together to the model dimension.

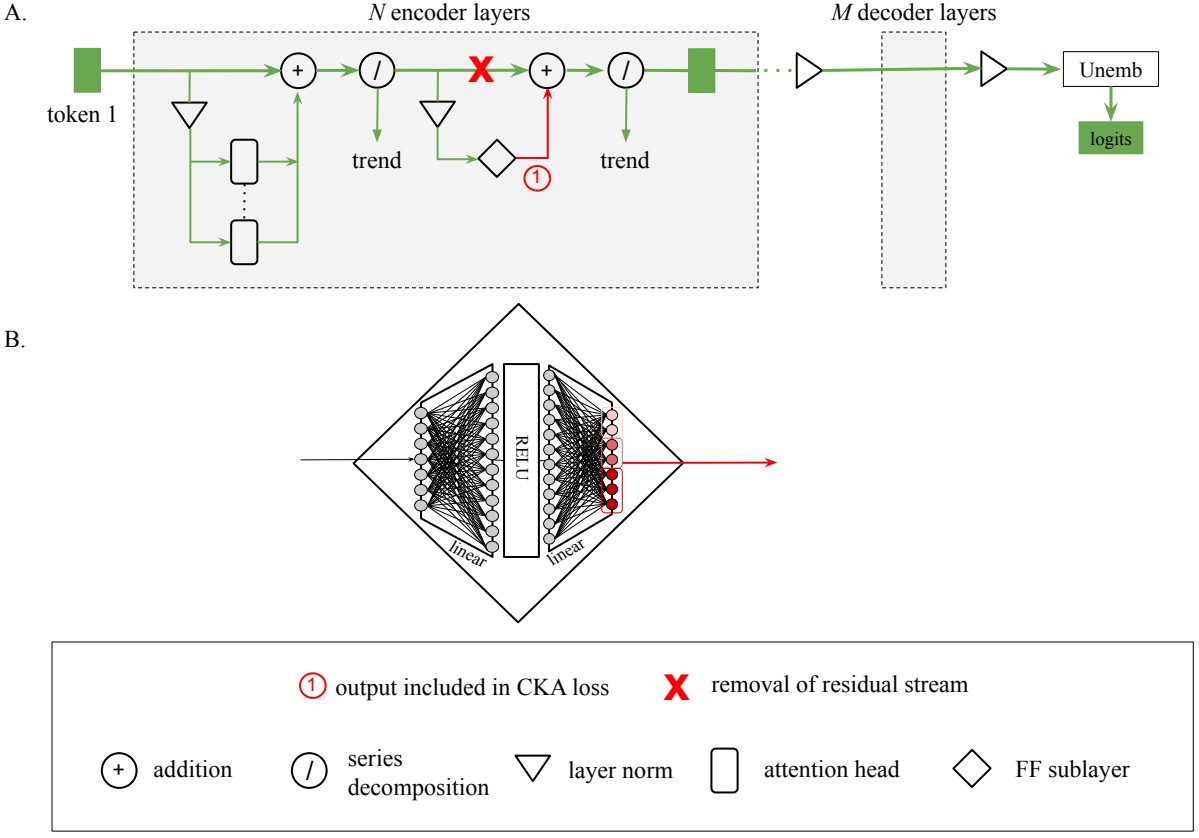

Figure 10: Architecture of Autoformer with a concept bottleneck in the feed-forward sublayer. Figure A shows the total architecture, including the removal of the residual stream at the location of the feed-forward sublayer. Figure B shows the overview of the feed-forward network consisting of two linear layers connected with the RELU activation function in between. The output of the final linear layer is used in the bottleneck. The output is split into three parts to assign the concepts to different components in the bottleneck.

# E   Detailed Overview Bottleneck Architecture

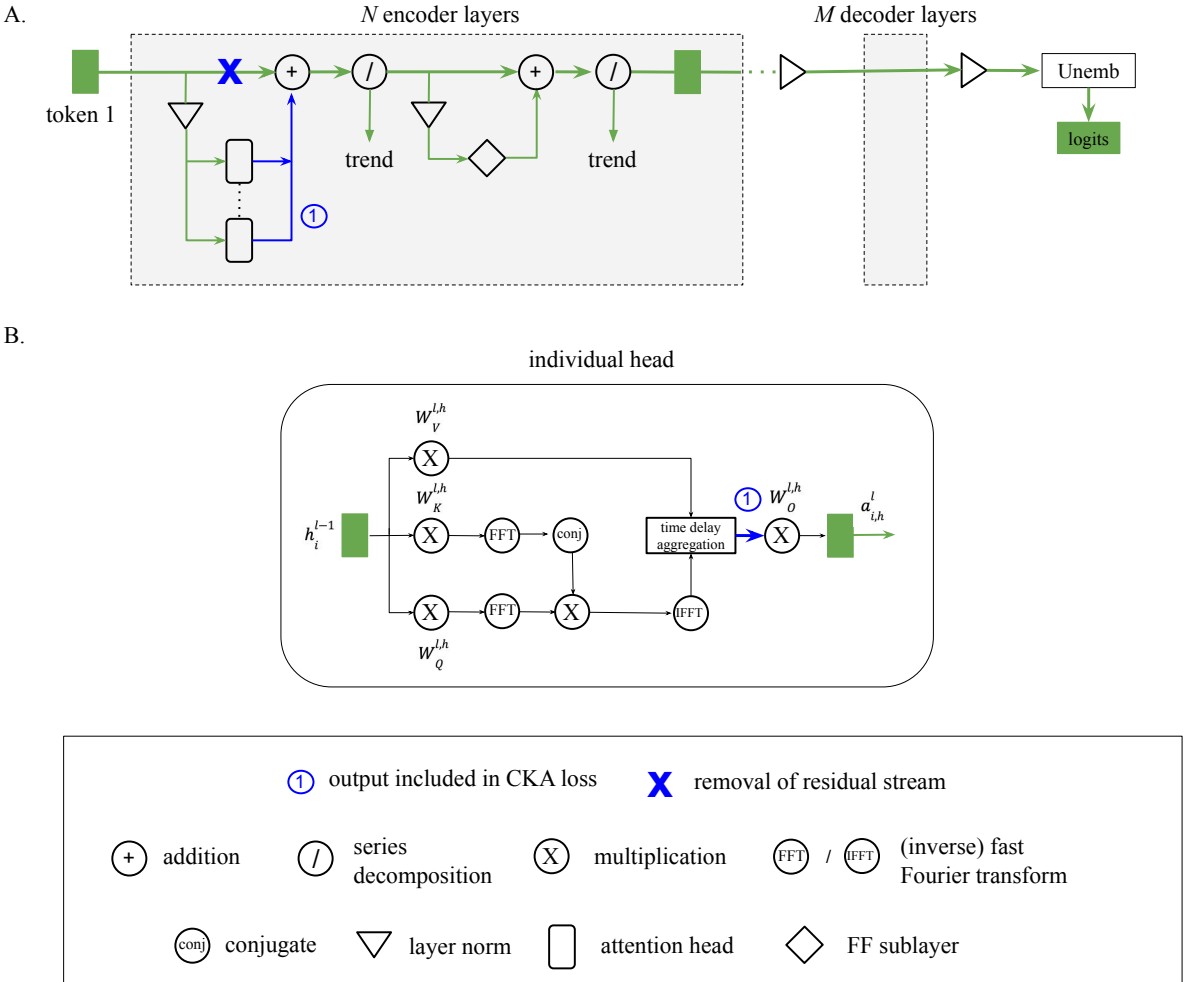

Figure 11: Architecture of Autoformer with a concept bottleneck in the attention mechanism. Figure A shows the total architecture, including the removal of the residual stream at the location of the attention block. Figure B shows the overview of an individual attention head $i$ with its input representation $h_i^{l-1}$ and output activation $a_{i,h}^l$. The representations after the time delay aggregation block are used in the bottleneck, because these correspond to the output of head $i$. Note that in practice, the individual head representations do not exist anymore after multiplication with the weights $W_o^{l,h}$, as this matrix projects the activations of all heads together to the model dimension.

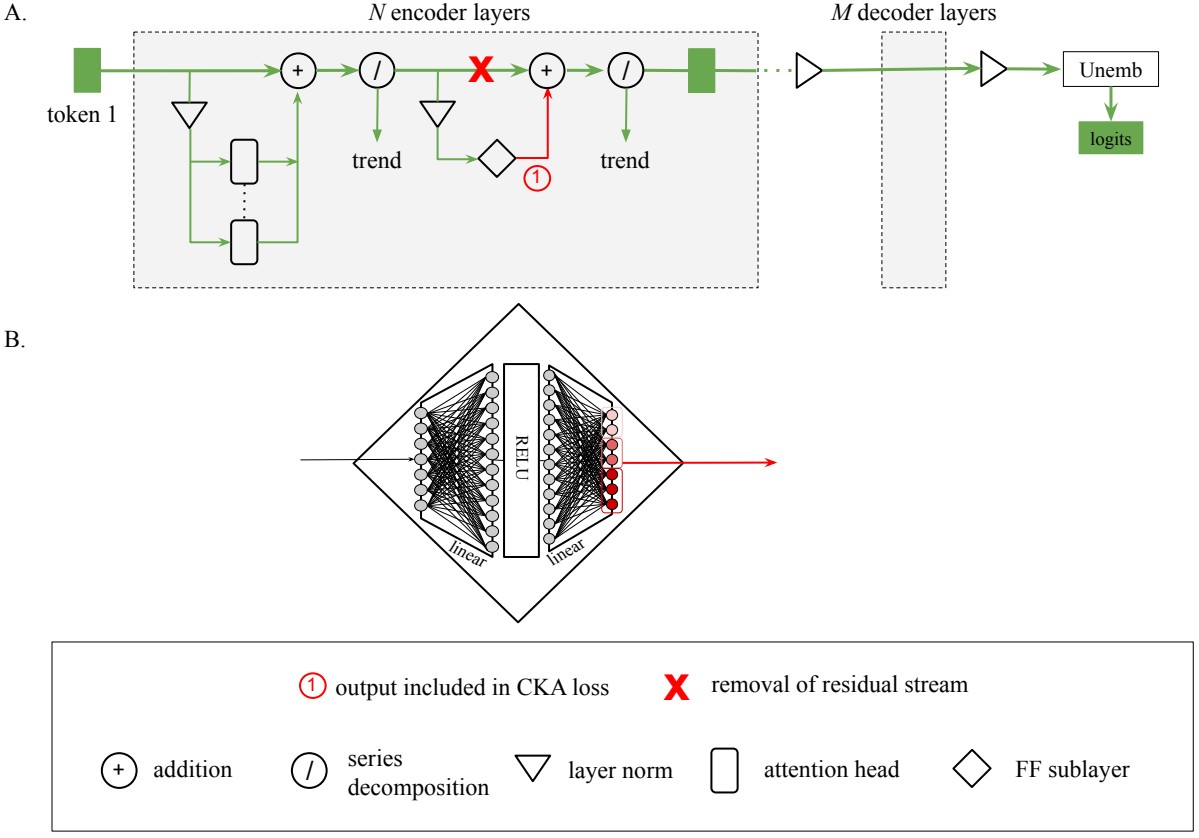

Figure 12: Architecture of Autoformer with a concept bottleneck in the feed-forward sublayer. Figure A shows the total architecture, including the removal of the residual stream at the location of the feed-forward sublayer. Figure B shows the overview of the feed-forward network consisting of two linear layers connected with the RELU activation function in between. The output of the final linear layer is used in the bottleneck. The output is split into three parts to assign the concepts to different components in the bottleneck.

## F Qualitative Results

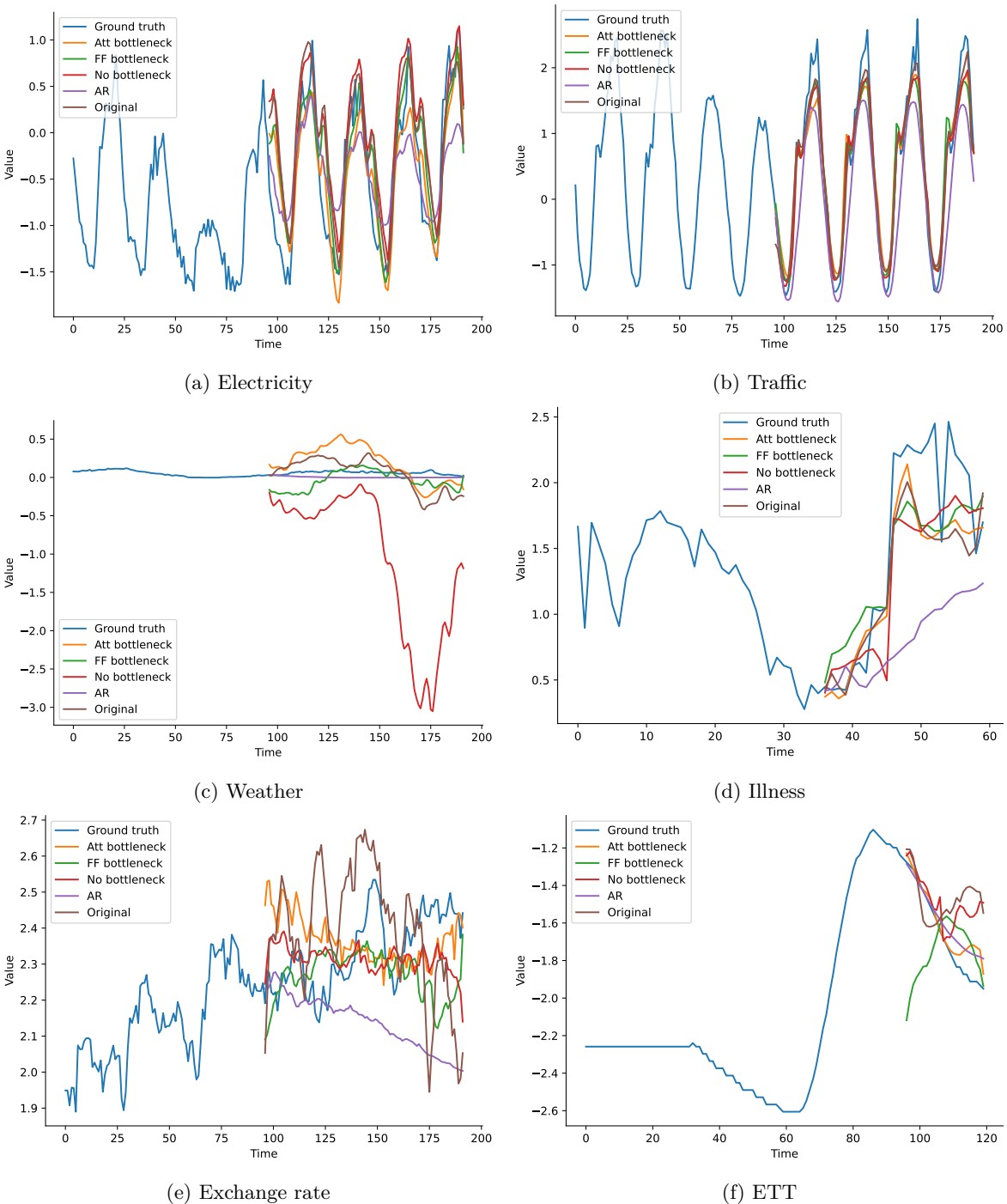

Figure 13: Forecasts on different datasets. The first part of the ground truth (shown in blue) is the input for the models, and the test set is used for each dataset.

# G  Detailed results

Table 3: Performance of different models in Mean Squared Error (MSE) and Mean Absolute Error (MAE). The bottlenecks **do** contain a free component ($c = 3$), and use AR as surrogate model. The model with no bottleneck is an original Autoformer of similar size. For all datasets, the shortest prediction lengths from Wu et al. (2021) are used, see Table 2. The standard deviation is determined using five different seeds.

| Free component | Att bottleneck | | FF bottleneck | | No bottleneck | | AR | | Wu et al. | |
|---|---|---|---|---|---|---|---|---|---|---|
| | MSE | MAE | MSE | MAE | MSE | MAE | MSE | MAE | MSE | MAE |
| Electricity | 0.231 ± 0.009 | 0.338 ± 0.005 | 0.207 ± 0.005 | 0.320 ± 0.005 | 0.280 ± 0.165 | 0.368 ± 0.111 | 0.497 | 0.522 | **0.201** ± 0.003 | **0.317** ± 0.004 |
| Traffic | 0.642 ± 0.022 | 0.393 ± 0.013 | **0.393** ± 0.013 | **0.377** ± 0.006 | 0.619 ± 0.015 | 0.387 ± 0.005 | 0.420 | 0.494 | 0.613 ± 0.028 | 0.388 ± 0.012 |
| Weather | 0.290 ± 0.027 | 0.354 ± 0.020 | 0.271 ± 0.016 | 0.341 ± 0.011 | 0.269 ± 0.000 | 0.344 ± 0.000 | **0.006** | **0.062** | 0.266 ± 0.007 | 0.336 ± 0.006 |
| Illness | 3.586 ± 0.241 | 1.313 ± 0.040 | 3.661 ± 0.237 | 1.322 ± 0.050 | 3.405 ± 0.208 | 1.295 ± 0.044 | **1.027** | **0.820** | 3.483 ± 0.107 | 1.287 ± 0.018 |
| Exchange rate | 0.195 ± 0.029 | 0.323 ± 0.025 | 0.155 ± 0.010 | 0.290 ± 0.013 | 0.152 ± 0.003 | 0.283 ± 0.003 | **0.082** | **0.230** | 0.197 ± 0.019 | 0.323 ± 0.012 |
| ETT | 0.177 ± 0.003 | 0.282 ± 0.004 | 0.174 ± 0.006 | 0.280 ± 0.005 | 0.155 ± 0.004 | 0.265 ± 0.002 | **0.034** | **0.117** | 0.255 ± 0.020 | 0.339 ± 0.020 |

Table 4: Performance on different datasets, where the bottlenecks **do not** contain a free component ($c = 2$). AR is used as surrogate model in the bottlenecks. The model with no bottleneck is an original Autoformer of similar size. For all datasets, the shortest prediction lengths from Wu et al. (2021) are used, see Table 2. The standard deviation is determined using five different seeds.

| No free component | Att bottleneck | | FF bottleneck | | No bottleneck | | AR | | Wu et al. | |
|---|---|---|---|---|---|---|---|---|---|---|
| | MSE | MAE | MSE | MAE | MSE | MAE | MSE | MAE | MSE | MAE |
| Electricity | 0.224 ± 0.006 | 0.332 ± 0.003 | 0.206 ± 0.009 | 0.321 ± 0.009 | 0.202 ± 0.006 | 0.318 ± 0.007 | 0.497 | 0.522 | **0.201** ± 0.003 | **0.317** ± 0.004 |
| Traffic | 0.629 ± 0.023 | 0.394 ± 0.015 | 0.627 ± 0.031 | 0.392 ± 0.025 | 0.613 ± 0.018 | **0.378** ± 0.007 | **0.420** | 0.494 | 0.613 ± 0.028 | 0.388 ± 0.012 |
| Weather | 0.281 ± 0.025 | 0.348 ± 0.018 | 0.260 ± 0.015 | 0.333 ± 0.013 | 0.257 ± 0.004 | 0.332 ± 0.005 | **0.006** | **0.062** | 0.266 ± 0.007 | 0.336 ± 0.006 |
| Illness | 3.966 ± 0.296 | 1.401 ± 0.073 | 3.721 ± 0.268 | 1.351 ± 0.053 | 3.585 ± 0.331 | 1.333 ± 0.070 | **1.027** | **0.820** | 3.483 ± 0.107 | 1.287 ± 0.018 |
| Exchange rate | 0.208 ± 0.026 | 0.333 ± 0.022 | 0.158 ± 0.009 | 0.293 ± 0.009 | 0.152 ± 0.006 | 0.284 ± 0.007 | **0.082** | **0.230** | 0.197 ± 0.019 | 0.323 ± 0.012 |
| ETT | 0.178 ± 0.011 | 0.283 ± 0.007 | 0.174 ± 0.01 | 0.283 ± 0.009 | 0.165 ± 0.004 | 0.274 ± 0.004 | **0.034** | **0.117** | 0.255 ± 0.020 | 0.339 ± 0.020 |

## H Hyper-Parameter Sensitivity

To verify the sensitivity to hyperparameter $\alpha$ in the loss function, we train the Autoformer with a feed-forward bottleneck on different values for $\alpha$, where the bottleneck contains a free component ($c = 3$) and the model is trained on the electricity dataset. The results are given in Figure 14. Interestingly, the error scores for all $\alpha < 1$ are close in value, which verifies that additionally training for interpretability does not hurt the performance, at least not in this set-up. Note that a low forecasting error cannot be expected for $\alpha = 1$, because in this edge case the loss function does not contain any term that represents the forecasting performance.

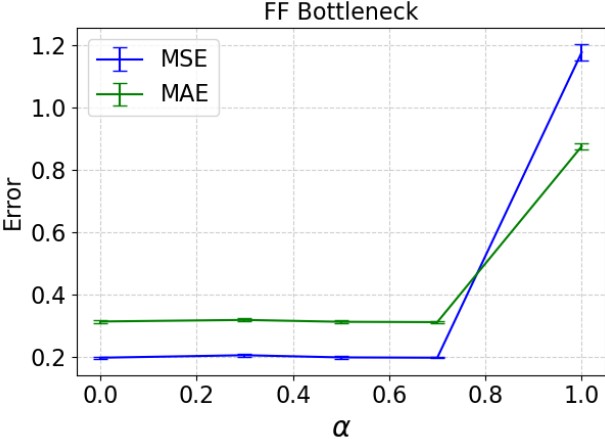

Figure 14: Performance of the Autoformer for different values of $\alpha$ in MSE and MAE.

Additionally, the CKA scores of the different models with the interpretable concepts (and other time features) are given in Figures 15, 16, and 17. Naturally, the CKA scores are the lowest in the setting $\alpha = 0$, and the scores from the bottleneck (layer1) increase over $\alpha$. Interestingly, the CKA scores from the bottleneck do not increase for higher values than $\alpha = 0.5$, although the scores of some other components do increase. This indicates that perfect similarity (i.e. CKA score of 1) to some interpretable concepts may not be reached.

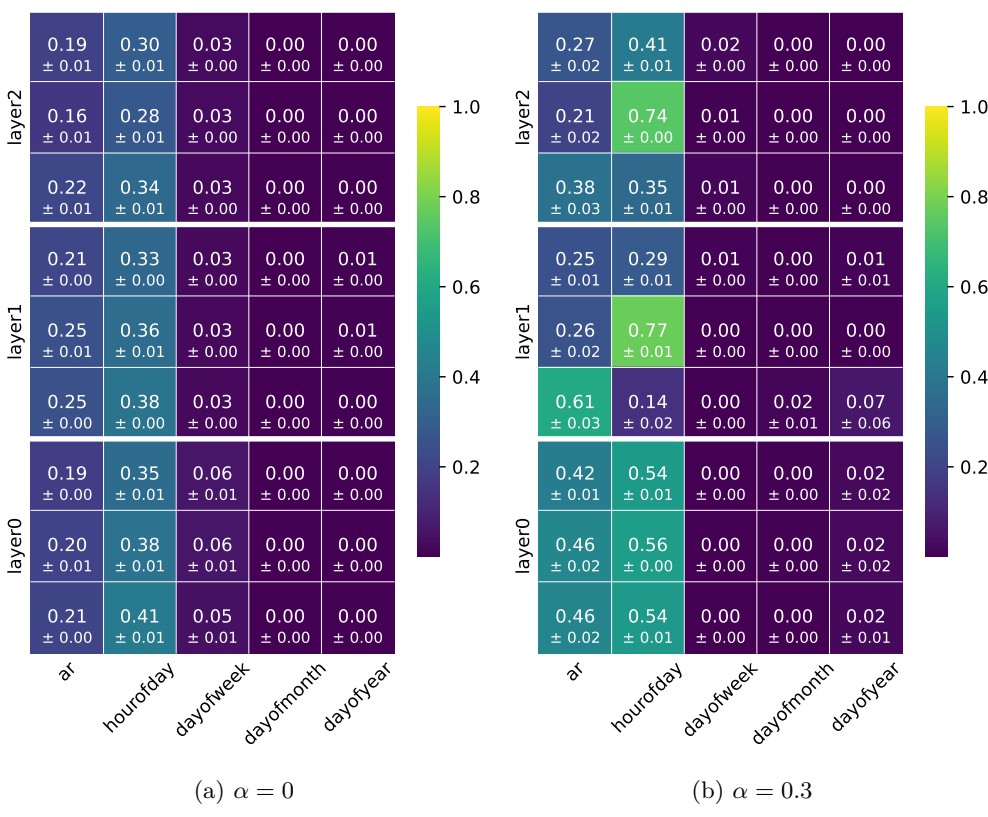

(a) $\alpha = 0$             (b) $\alpha = 0.3$

Figure 15: CKA scores of the feed-forward bottleneck Autoformer on electricity data for different values of hyperparameter $\alpha$. The scores are calculated using three batches of size 32 of the test data set.

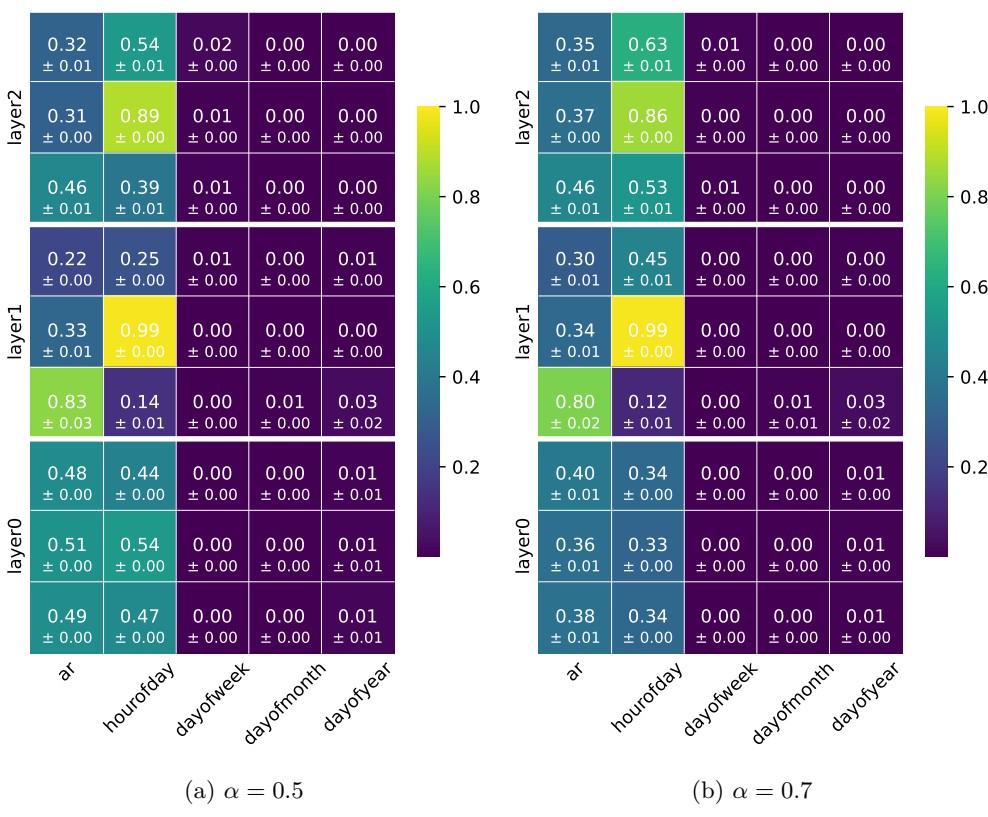

(a) $\alpha = 0.5$        (b) $\alpha = 0.7$

Figure 16: CKA scores of the feed-forward bottleneck Autoformer on electricity data for different values of hyperparameter $\alpha$. The scores are calculated using three batches of size 32 of the test data set.

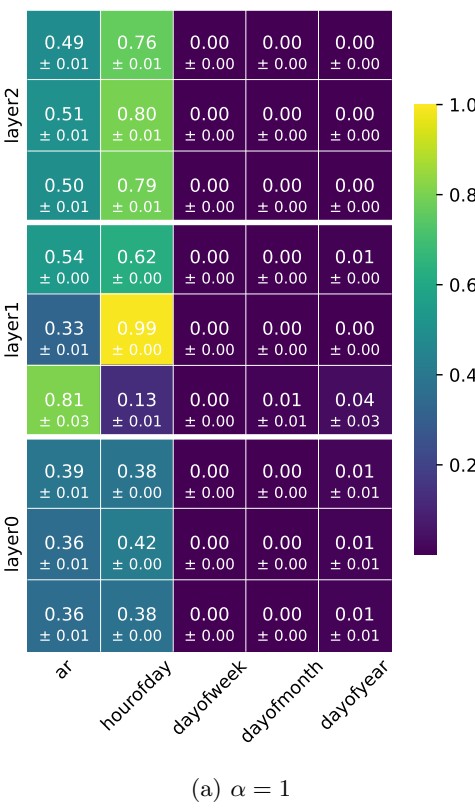

(a) $\alpha = 1$

Figure 17: CKA scores of the feed-forward bottleneck Autoformer on electricity data for hyperparameter $\alpha = 1$. The scores are calculated using three batches of size 32 of the test data set.

# I CKA Analysis for More Datasets

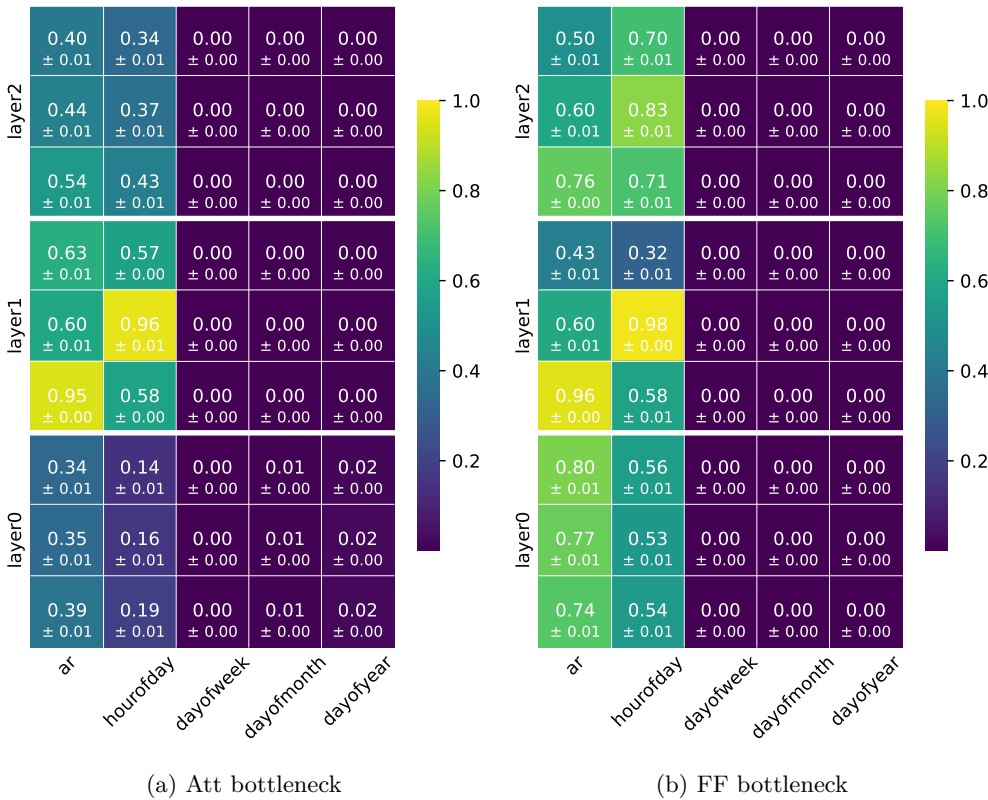

(a) Att bottleneck

(b) FF bottleneck

Figure 18: Traffic - CKA scores of the attention bottleneck Autoformer on traffic data. The scores are calculated using three batches of size 32 of the test data set.

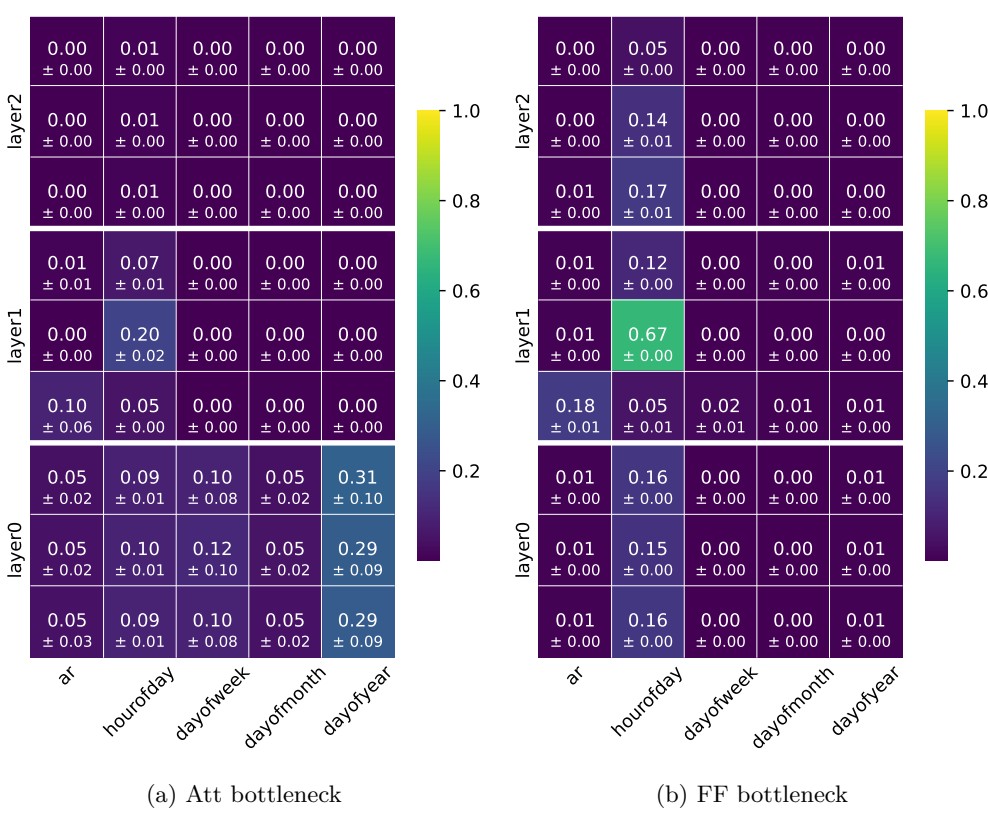

(a) Att bottleneck  (b) FF bottleneck

Figure 19: Weather - CKA scores of the attention bottleneck Autoformer on weather data. The scores are calculated using three batches of size 32 of the test data set.

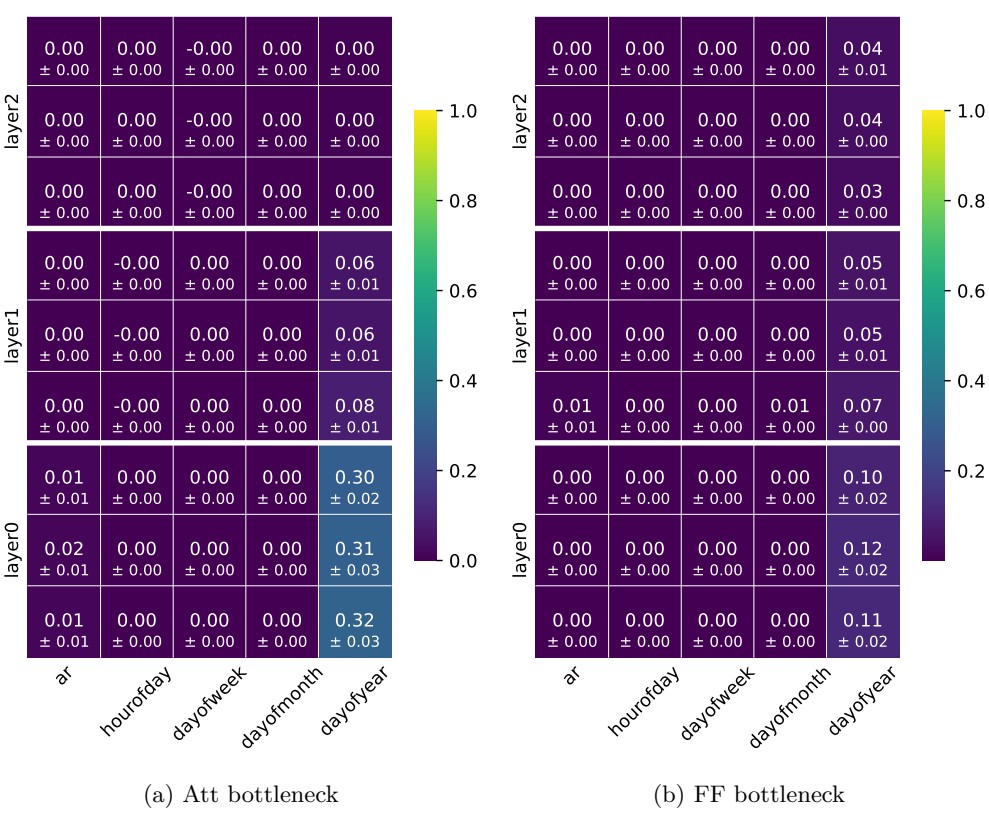

(a) Att bottleneck

(b) FF bottleneck

Figure 20: Illness - CKA scores of the attention bottleneck Autoformer on the illness data set. The scores are calculated using three batches of size 32 of the test data set.

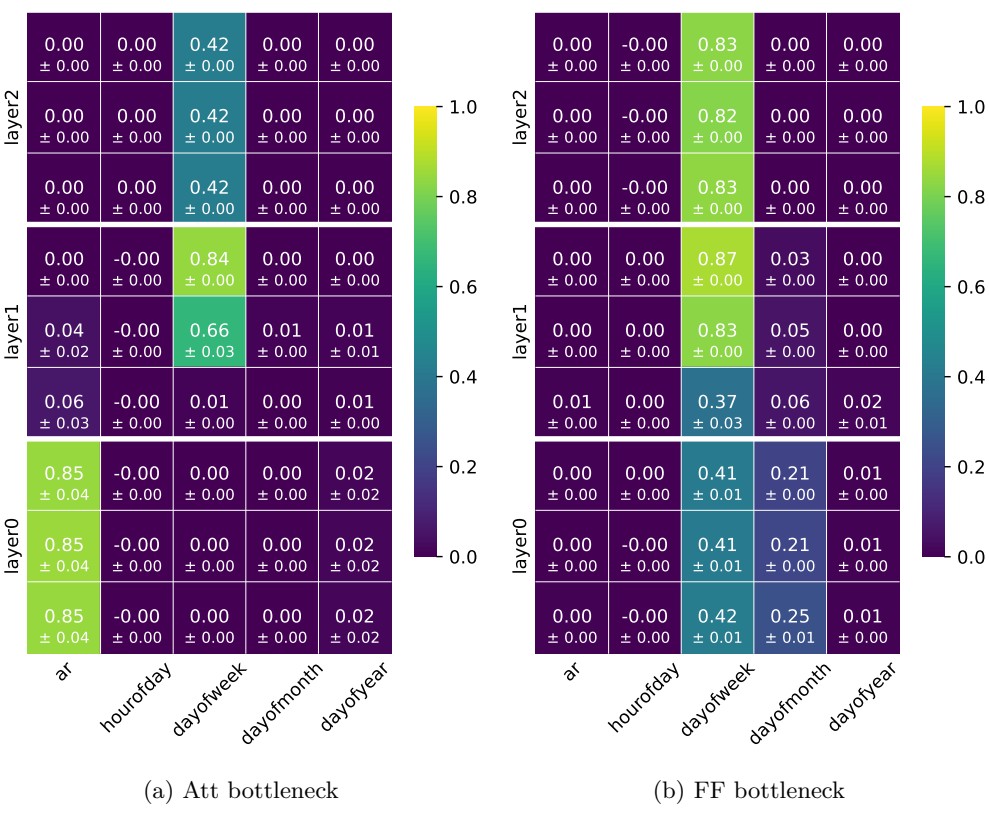

(a) Att bottleneck         (b) FF bottleneck

Figure 21: Exchange rate - CKA scores of the attention bottleneck Autoformer on the exchange rate data set. The scores are calculated using three batches of size 32 of the test data set.

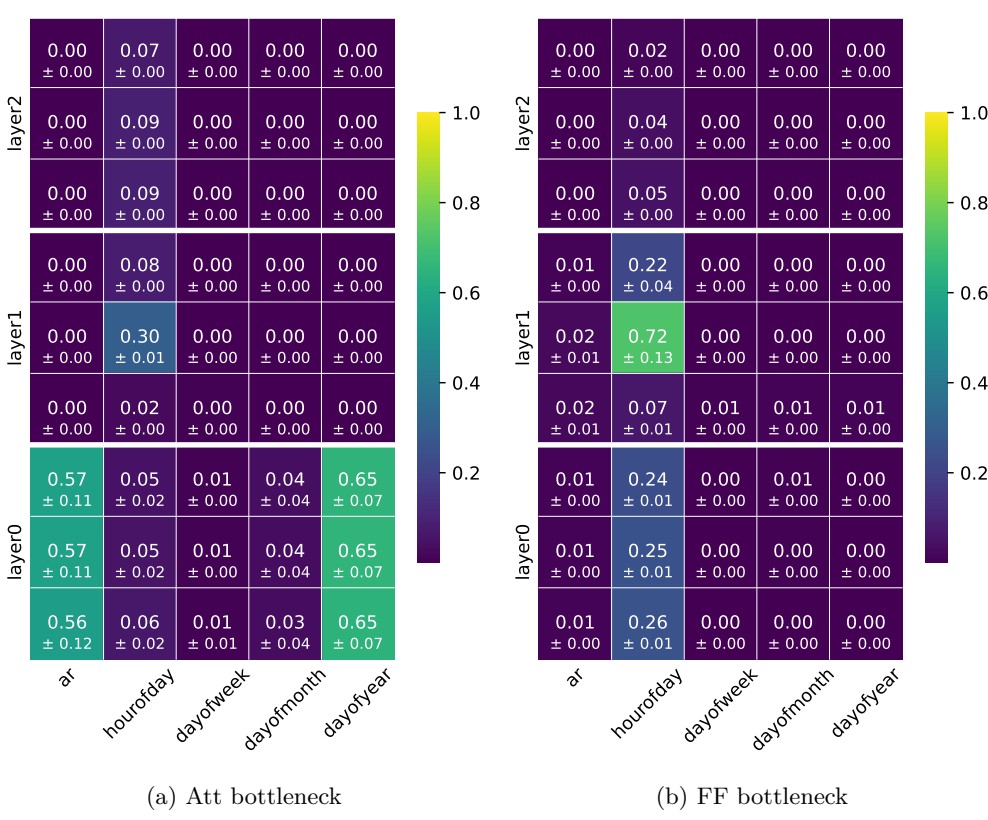

(a) Att bottleneck

(b) FF bottleneck

Figure 22: ETT - CKA scores of the attention bottleneck Autoformer on the ETT data set. The scores are calculated using three batches of size 32 of the test data set.

# J  Application of Framework to Vanilla Transformer

To demonstrate the generality of the concept bottleneck framework, we apply it to an additional Transformer architecture, namely the *vanilla Transformer* (the original architecture from which all Transformer models, including all time series Transformers, are derived). We train it using the same six benchmark datasets and perform a similar, but less extensive, analysis as done for the Autoformer model. Note that the architecture of the Transformer is *not* modified, and the timestamps are included as an embedding (in addition to the positional embedding).

## J.1  Performance Analysis

The performance of the vanilla Transformer model with and without bottleneck is given in Table 5. We train the bottleneck with a 'free' component (the side channel), i.e., with $c = 3$. Note that Wu et al. (2021) do not provide scores for these benchmark forecasting datasets, therefore we cannot include them in the table. The results show that the vanilla Transformer performs, unsurprisingly, worse than the Autoformer, and for most datasets also worse than the linear AR model. However, most relevant, for our purposes, is that across the datasets using a concept bottleneck does not hurt the overall performance of the vanilla Transformer.

Table 5: Performance of different vanilla Transformer models. For both metrics, it holds that a lower score indicates a better performance, where the best results are **bold**, and the second-best are underlined.

|  | Att bottleneck | | FF bottleneck | | No bottleneck | | AR | |
|---|---|---|---|---|---|---|---|---|
|  | MSE | MAE | MSE | MAE | MSE | MAE | MSE | MAE |
| Electricity | 0.275 | 0.371 | **0.268** | **0.362** | 0.275 | 0.371 | 0.497 | 0.522 |
| Traffic | 0.708 | 0.394 | 0.703 | 0.397 | 0.684 | **0.376** | **0.420** | 0.494 |
| Weather | 0.400 | 0.450 | 0.381 | 0.410 | 0.362 | 0.415 | **0.006** | **0.062** |
| Illness | 3.380 | 1.280 | 3.323 | 1.252 | 3.321 | 1.273 | **1.027** | **0.820** |
| Exchange rate | 0.675 | 0.642 | 0.677 | 0.633 | 0.694 | 0.662 | **0.082** | **0.230** |
| ETT | 0.230 | 0.328 | 0.185 | 0.299 | 0.166 | 0.294 | **0.034** | **0.117** |

## J.2  CKA Analysis

After training the vanilla Transformer with the bottleneck framework, we evaluate the similarity of its hidden representations to the interpretable concepts using CKA, see Figure 23. Recall that CKA scores are defined in the range from 0 to 1, where 1 indicates perfect similarity. Both components in the two types of bottleneck show very high similarity to their target concept. Interestingly, the first component in the bottleneck (the AR concept) shows a higher similarity to the AR representations than the Autoformer (see Figure 4), presumably because the decomposition structure of the Autoformer hinders learning a linear function.

## J.3  Component Visualizations

We visualize the contributions of each component in the bottleneck using the Decoder Lens method (Langedijk et al., 2023), see Figure 24. We obtain the output from each component individually by masking the other components with zero (close to the mean). Each component seems to provide similar contributions to the forecast as their respective counterpart in the Autoformer model. In particular, the first component (see Figure 24a) produces forecasts of correct seasonality and few irregularities, similar to the AR model. The second component (see Figure 24b) follows the hour-of-day feature, and the free head (see Figure 24c) picks up on high-frequency data patterns.

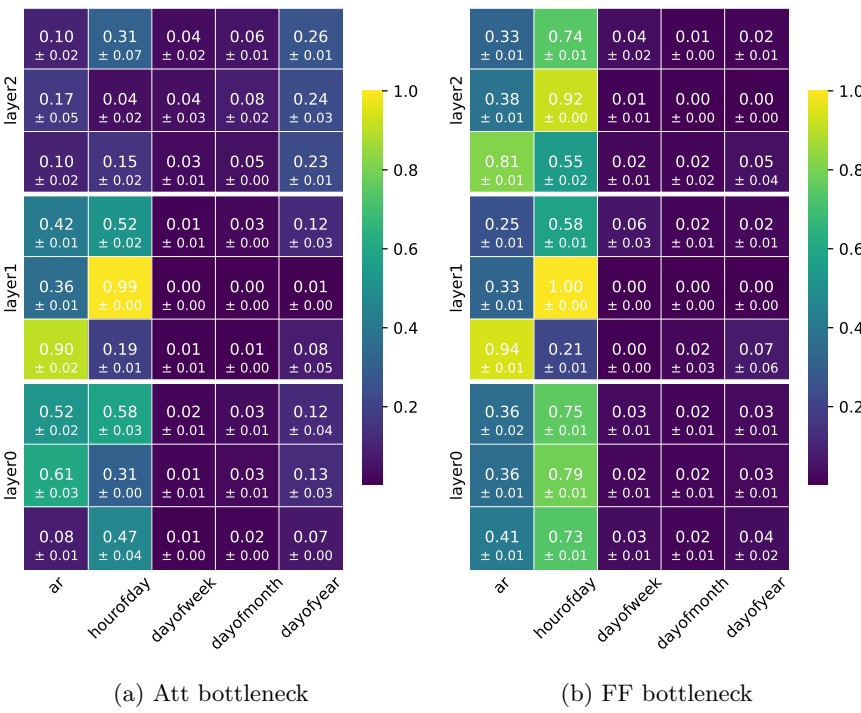

(a) Att bottleneck       (b) FF bottleneck

Figure 23: CKA scores of the vanilla Transformer's encoder (containing three heads per layer) from the attention and feed-forward bottleneck on the electricity dataset, where each score denotes the similarity of an individual component. The first component of `layer1` is trained to be similar to AR, and the second component to the hour-of-day concept (lower and middle row in the figure, respectively). The scores are calculated using three batches of size 32 from the test data set.

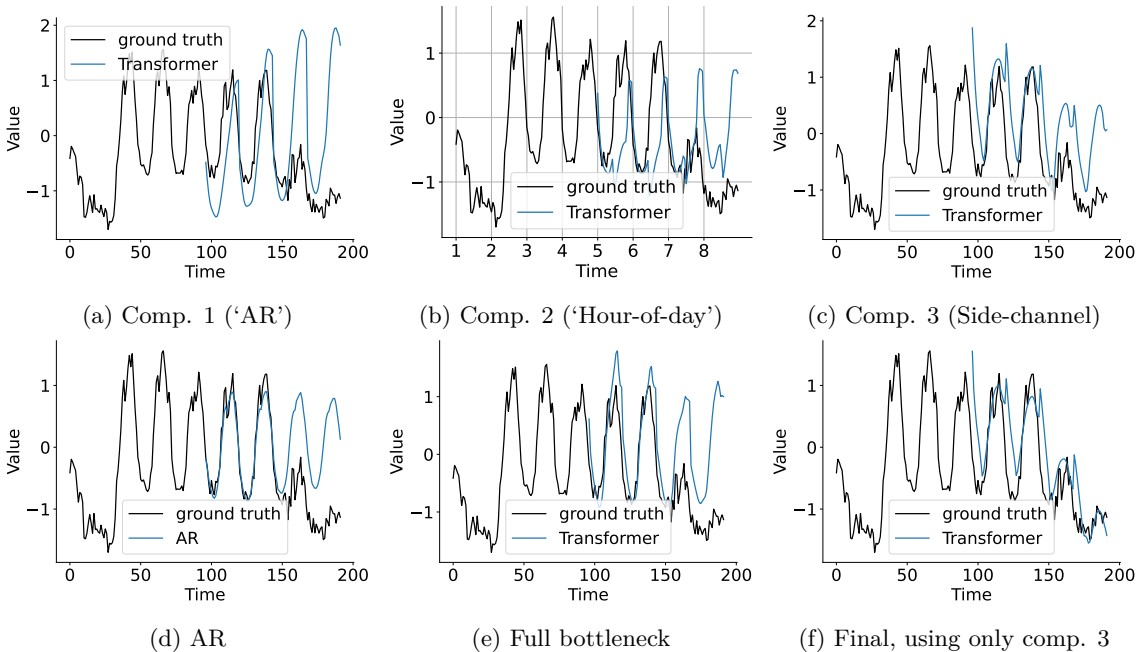

(a) Comp. 1 ('AR')          (b) Comp. 2 ('Hour-of-day')          (c) Comp. 3 (Side-channel)

(d) AR          (e) Full bottleneck          (f) Final, using only comp. 3

Figure 24: Vanilla Transformer forecasts from the components in the bottleneck layer (FF bottleneck on electricity data) in 24a, 24b and 24c. They are obtained by masking the other components with zero (the mean). The first half of the ground truth forms the input to the model. Note that the horizontal axes are the same across all figures, but Figure 24b contains a grid of days instead of numbered hours. Figure 24d shows the forecast made by the surrogate model AR; Figure 24e shows the forecast of the entire layer (i.e., all components together), and 24f shows the forecast of the final layer when only the third component is used in the bottleneck layer. Note the difference between Figures 24c and 24f, where we decode from the bottleneck and the final layer, respectively.

## J.4  Intervention

We perform the intervention experiment in the same set-up as for the Autoformer model. That is, we delay the input timestamps with a fixed number of hours to obtain shifted timestamps, and perform an intervention in the bottleneck by substituting the activations based on the shifted time with the activations from the original time. We use a vanilla Transformer trained on the electricity dataset, and perform shifts of up to and including 23 hours. We compare the performance of the intervention with out-of-the-box performance of the same model on the shifted dataset. The results are shown in Figure 25. For both types of bottlenecks, the intervention performs best for all timeshifts, by keeping the error scores marginally close to the original performance (with no timeshift). This indicates that the model effectively learns to represent the hour-of-day concept in the dedicated head, which is able to provide control over the model's behavior.

## J.5  Conclusion

By repeating the set of experiments for the vanilla Transformer model, we provided further evidence for the generality of the concept bottleneck framework. In particular, we showed that the framework can be applied to the vanilla Transformer model, without having any significant impact on the overall model performance, while providing improved interpretability.

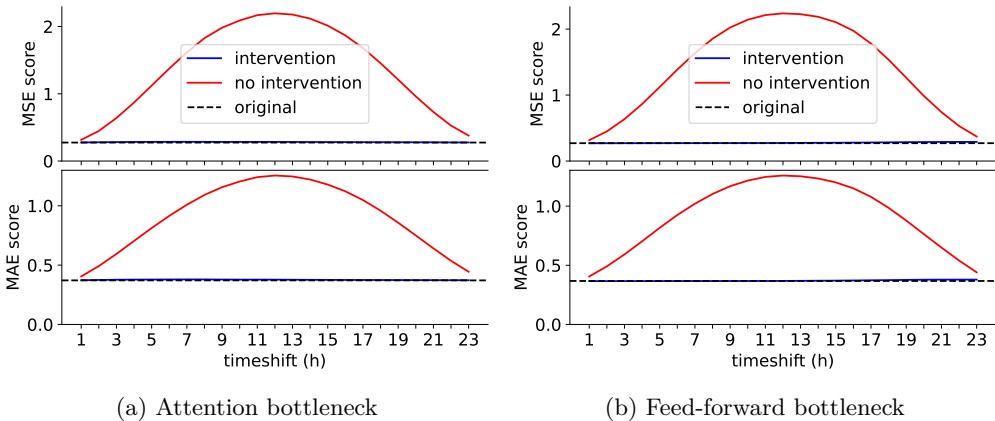

(a) Attention bottleneck          (b) Feed-forward bottleneck

Figure 25: Performance of the bottleneck vanilla Transformer on electricity data with shifted timestamps. The dashed line represents the performance of the same model on the original data, i.e., with no timeshift.

## K   Application of Framework to FEDformer

To demonstrate the generality of our concept bottleneck framework, we apply it to *FEDformer* (Zhou et al., 2022). This is a Transformer architecture containing *Fourier enhanced blocks* and *wavelet enhanced blocks* to represent time series in the frequency domain. For more details, we refer to the original authors Zhou et al. (2022). We train the model on the same six datasets and perform an interpretability analysis.

### K.1   Performance Analysis

The performance of the FEDformer with and without bottleneck is given in Table 6. We train the bottleneck with a 'free' component (the side channel), i.e., with $c = 3$. Note that the model by Zhou et al. (2022) is of a different size (two encoder layers with eight heads per layer). Interestingly, we find for some datasets (e.g. electricity and illness) that including a bottleneck increases the performance, while it has little effect on the performance for the other datasets. We can conclude for all datasets that including a bottleneck does not hurt performance.

Table 6: Performance of FEDformer. For both metrics, it holds that a lower score indicates a better performance, where the best results are **bold**, and the second-best are underlined.

|  | Att bottleneck | | FF bottleneck | | No bottleneck | | AR | | Zhou et al. | |
|---|---|---|---|---|---|---|---|---|---|---|
|  | MSE | MAE | MSE | MAE | MSE | MAE | MSE | MAE | MSE | MAE |
| Electricity | **0.185** | **0.302** | 0.186 | 0.303 | 0.189 | 0.304 | 0.497 | 0.522 | 0.193 | 0.308 |
| Traffic | 0.585 | 0.364 | 0.585 | 0.364 | 0.573 | **0.358** | **0.420** | 0.494 | 0.587 | 0.366 |
| Weather | 0.221 | 0.299 | 0.219 | 0.296 | 0.334 | 0.397 | **0.006** | **0.062** | 0.217 | 0.296 |
| Illness | 3.070 | 1.217 | 3.076 | 1.219 | 3.111 | 1.232 | **1.027** | **0.820** | 3.228 | 1.260 |
| Exchange rate | 0.147 | 0.277 | 0.145 | 0.275 | 0.146 | 0.276 | **0.082** | **0.230** | 0.148 | 0.278 |
| ETT | 0.079 | 0.193 | 0.079 | 0.192 | 0.077 | 0.190 | **0.034** | **0.117** | 0.203 | 0.287 |

### K.2   CKA Analysis

After training the FEDformer with our concept bottleneck framework, we evaluate the similarity of the hidden representations to the interpretable concepts using CKA, see Figure 26. Recall that CKA scores are defined in the range from 0 to 1, where 1 indicates perfect similarity. Both components in the two types of bottleneck show a very high similarity to their target concept, indicating a successful training on interpretability.

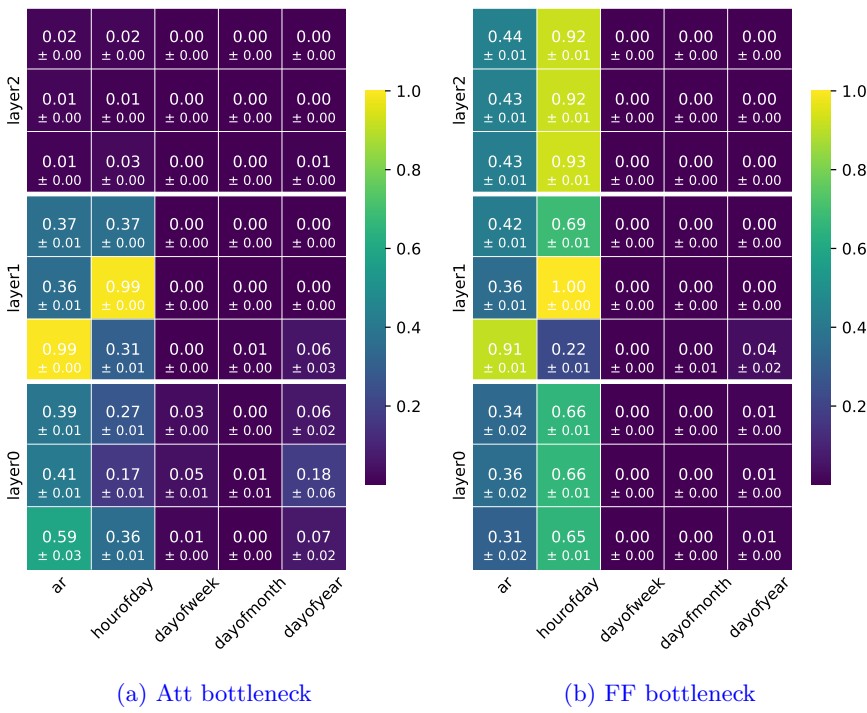

(a) Att bottleneck        (b) FF bottleneck

Figure 26: CKA scores of the FEDformer's encoder (containing three heads per layer) from the attention and feed-forward bottleneck on the electricity dataset, where each score denotes the similarity of an individual component. The first component of `layer1` is trained to be similar to AR, and the second component to the hour-of-day concept (lower and middle row in the figure, respectively). The scores are calculated using three batches of size 32 from the test data set.

### K.3  Intervention

Additionally, we perform the intervention experiment in the same set-up as for the other Transformer models. That is, we delay the input timestamps with a fixed number of hours and perform an intervention in the bottleneck by substituting the activations with those based on the original time. We compare the performance of the intervention with out-of-the-box performance of the same model on the shifted dataset. The results are shown in Figure 27. For both types of bottlenecks, the intervention performs best for all timeshifts, by keeping the error marginally close to the original performance (without timeshift). This indicates that the model effectively learns to represent the hour-of-day concept in the dedicated head, which is able to provide control over the model's behavior.

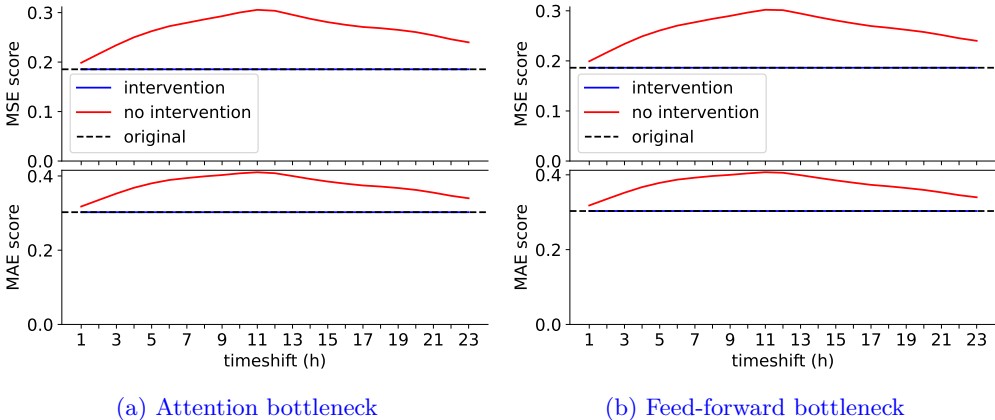

(a) Attention bottleneck          (b) Feed-forward bottleneck

Figure 27: Performance of the bottleneck FEDformer on electricity data with shifted timestamps. The dashed line represents the performance of the same model on the original data, i.e., with no timeshift.

### K.4  Conclusion

By repeating the set of experiments for the FEDformer model, we provided further evidence for the generality of the concept bottleneck framework. In particular, we showed that the framework can be applied to the FEDformer model, without having any significant impact on the overall model performance, while providing improved interpretability.

## L  Synthetic Data

To increase the understanding of how the concepts in the bottleneck can be leveraged, we train the model on a synthetic dataset.

### L.1  Dataset

We generate a synthetic time series as the sum of different functions. In particular, the dataset is generated using the function $f_{Total}$ with time $t$ as follows:

$$f_{Total}(t) = f_1(t) + f_2(t) + f_3(t),$$

where:

$$f_1(t) = \sin(2\pi t),$$
$$f_2(t) = \frac{1}{2}\sin(4\pi t + \frac{\pi}{4}),$$
$$f_3(t) = \frac{1}{4}\sin(6\pi t + \frac{\pi}{2}) + \epsilon_t.$$

Note that all functions $f_1, f_2$ and $f_3$ follow a periodic structure, and $f_3$ contains random noise $\epsilon$ from a normal distribution with standard deviation of 0.2. See Figure 28 for a visualization of the functions.

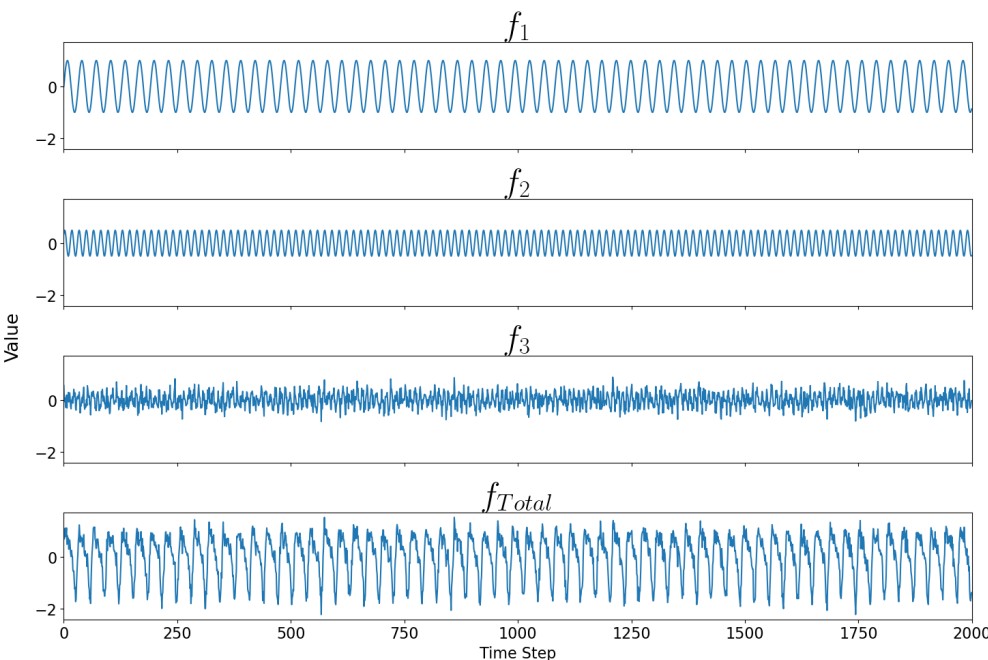

Figure 28: The synthetic time series dataset.

### L.2  Experiment and Results

We train an Autoformer model on the synthetic dataset using the concept bottleneck framework. Each concept in the bottleneck is defined as one of the underlying functions (i.e., $f_1$, $f_2$ or $f_3$), for which the ground-truth is known by construction. The model contains three encoder layers, with three attention heads per layer. We apply the bottleneck to the attention heads of the second encoder layer. Additionally, we train the bottleneck using different values for hyperparameter $\alpha$, which controls the weight of the CKA loss in the total loss function (see Section 3.1).

As expected, we find for all values $\alpha < 1$ that the model is able to forecast the dataset well, see Figure 29. Note that a low forecasting error cannot be expected for $\alpha = 1$, because in this edge case the loss function does not contain any term that represents the forecasting error. Remarkably, for all other cases, the performance of the Autoformer seems to improve as $\alpha$ increases. This suggests that properly chosen concepts improve the performance of the model, at least when the ground-truth underlying functions are known. It should be noted that the standard deviation is higher for all $\alpha > 0$, which indicates that initialization of the parameters is important when learning the bottleneck. Additionally, visualizations of the predictions are given in Figure 30.

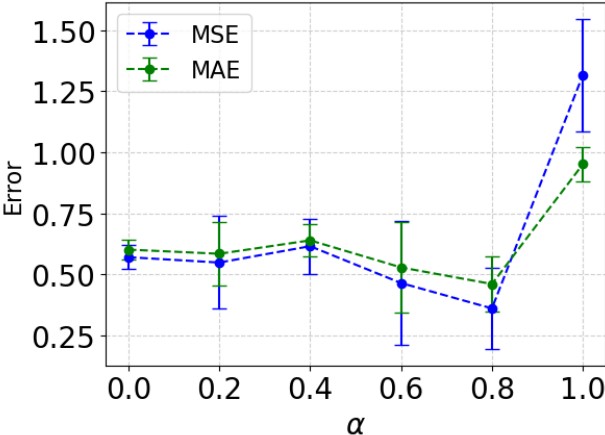

Figure 29: Performance on the synthetic dataset for different values of $\alpha$, using an Autoformer with attention bottleneck. For both metrics, it holds that a lower score indicates a better performance. The standard deviation is provided over 5 different seeds.

Additionally, the different values of hyperparameter $\alpha$ show clearly how the different concepts are leveraged by the model, see Figure 31. The figure shows the similarity scores between the attention heads and the different underlying functions of the dataset. Without the CKA loss, at $\alpha = 0$, the different heads in `layer1` of the model do not show high similarity to their respective concepts, i.e., functions. Instead, all heads have a high similarity to concept $f_2$. This is different for higher values of $\alpha$, where the different heads show higher similarity to their respective concepts. Note that the third concept $f_3$ cannot be perfectly learned by the model because of the random noise component.

All in all, these results show that a higher value for $\alpha$, which is equivalent to a higher weight of the CKA loss in the total loss function, results in more similarity of the bottleneck components to their respective concepts, as expected.

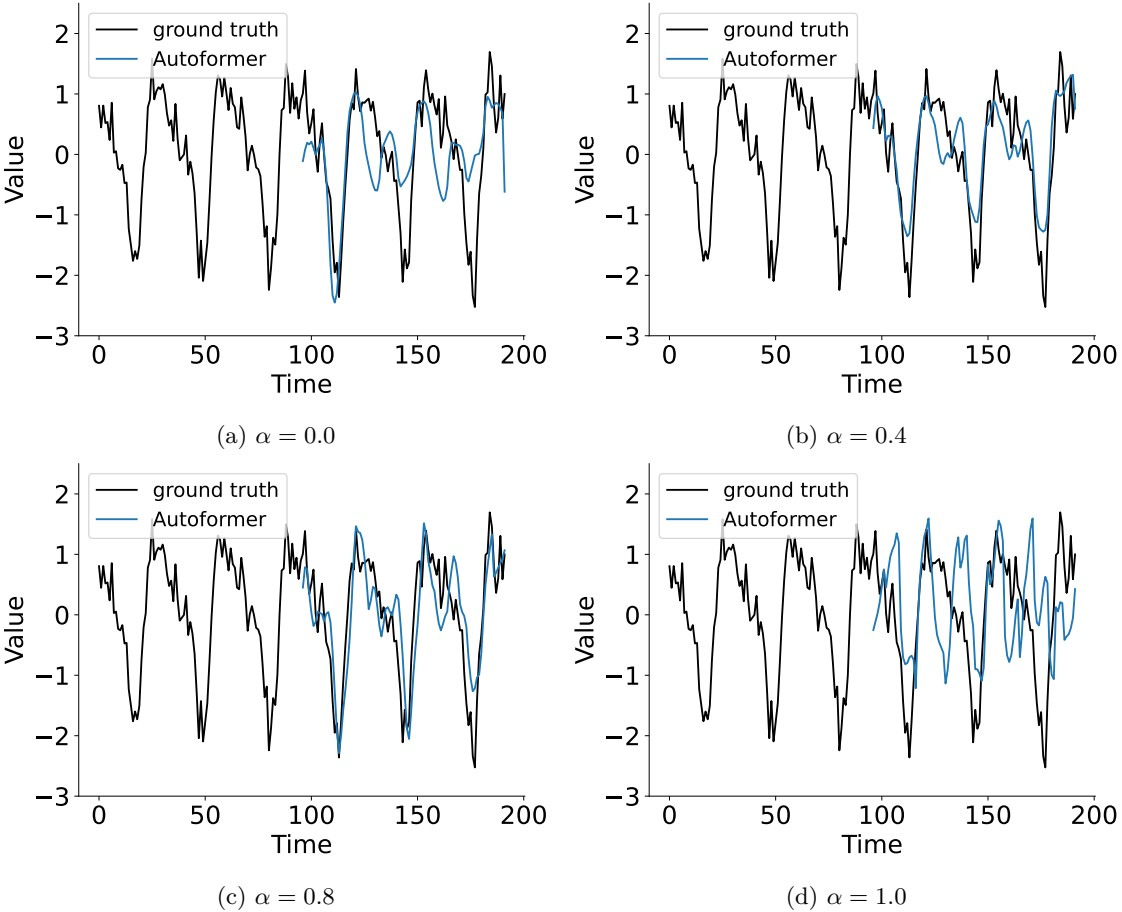

Figure 30: Predictions of the Autoformer model on a sample from the test dataset. The Autoformer is trained with an attention bottleneck using different values of hyperparameter $\alpha$ and the same seed.

## M Effect of AR as Surrogate Model

Interestingly, the AR model outperforms the Autoformer for some datasets (see Table 1). This raises the question whether the AR surrogate model makes up for any loss in performance introduced by the concept bottleneck.

To test this, we train an Autoformer without the AR concept. Specifically, we include the time concept and a free component in the feed-forward bottleneck. Here, the free component refers to a component in the bottleneck that is not included in the CKA loss (see Section 3.2).

The performance on the electricity data for this model is (MSE: 0.206, MAE: 0.321), which is seemingly identical to the original performance of (MSE: 0.207, MAE: 0.320). This suggests that it is not the AR head that makes up for the loss in performance. The CKA plots, see Figure 32, verify that there is no component in the minimal set-up (without AR) that is very similar to the AR model, unlike in the original set-up. So, these results show that the AR model does not add performance to the bottleneck model, merely interpretability.

Additionally, we refer the reader to Appendix L, where we perform more experiments on training the bottleneck without the AR surrogate model.

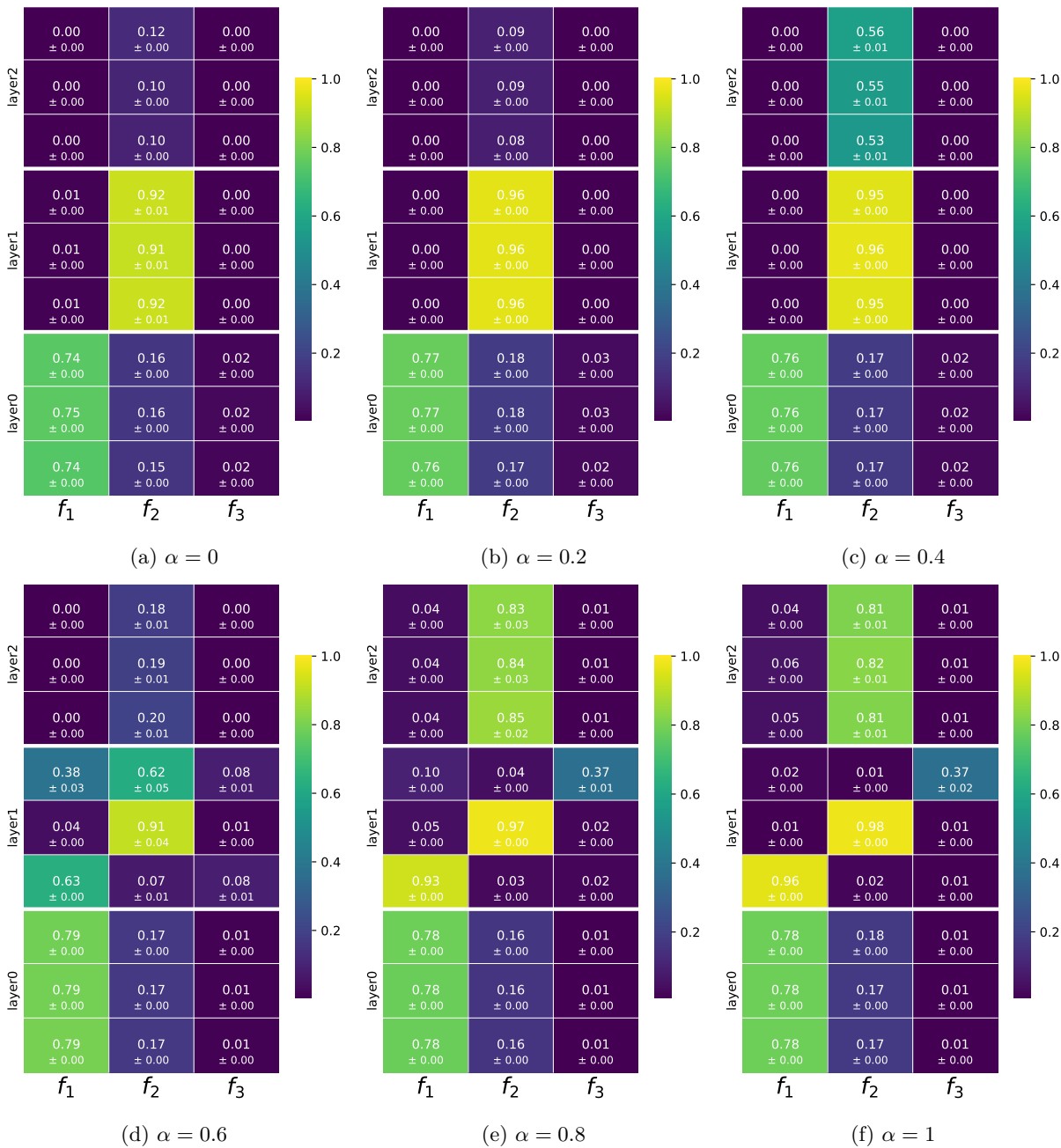

Figure 31: CKA scores of the attention bottleneck Autoformer on synthetic data for different values of hyperparameter $\alpha$. The scores are calculated using three batches of size 32 of the test data set.

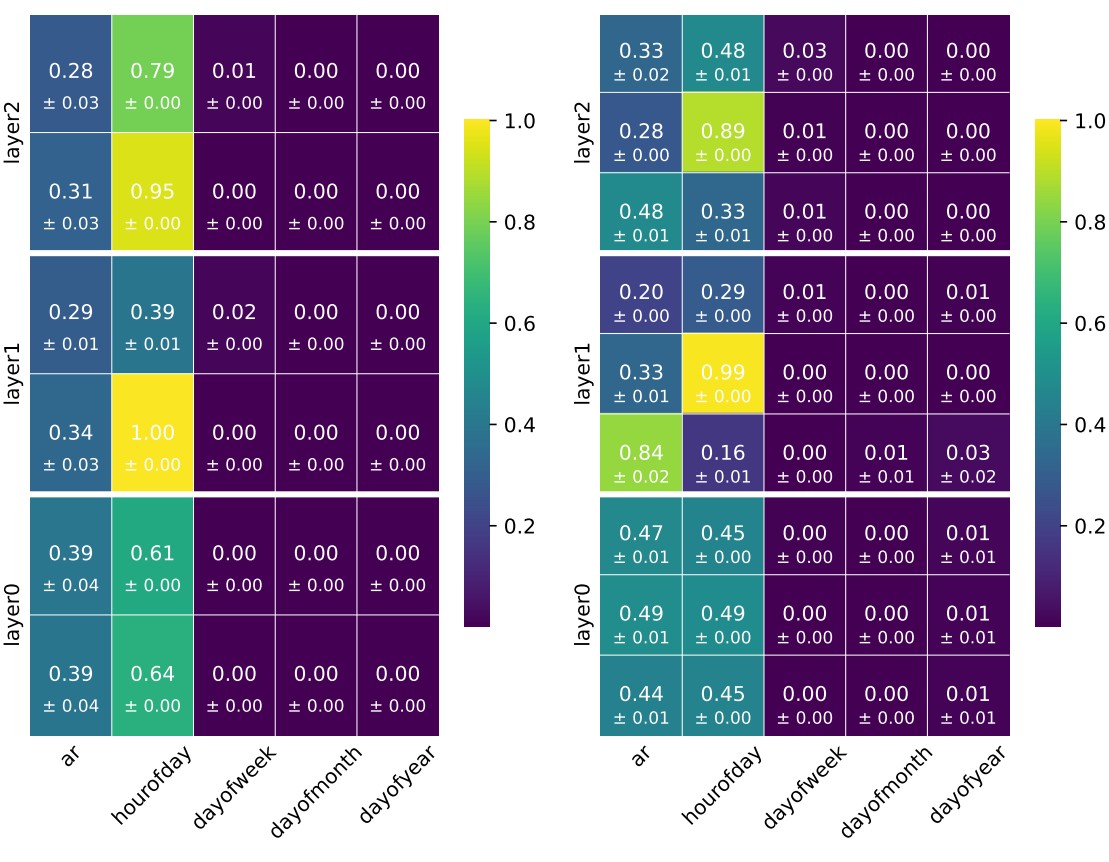

(a) Without AR (MSE: 0.206, MAE: 0.321)  (b) With AR (MSE: 0.207, MAE: 0.320)

Figure 32: CKA plots of two Autoformer models with feed-forward bottlenecks. The model in 32a is trained without AR in the bottleneck, while the model in 32b is trained with AR. Note that the upper component in `layer1` is the free component in both plots.

# N    Removal of Timestamps from Input of Transformers

In general, time-series transformers take sequential data values $X$ and timestamps $T$ as input. Therefore, a bottleneck trained on timestamp information could copy this concept from the input, instead of learning it from the time series values. To assess the learning abilities of novel information in the bottleneck layer, we train an Autoformer without timestamps, but include a bottleneck component to be similar to the hour-of-day feature (without providing this as input).

## N.1    Performance

In particular, we set timestamps $T$ to be a zero vector in this experiment, but keep the hour-of-day feature $T_{hourofday}$ to train our bottleneck. The results are shown in Table 7. For most datasets, training without timestamps hurts performance, as expected. Interestingly, it varies by dataset whether including a bottleneck increases performance. For instance, bottleneck models on the weather data outperform the original model without bottleneck, but hurt performance for the weather data. Overall claims on the effect of the bottleneck cannot be made based on these results.

Table 7: Performance of Autoformer trained without timestamps (rightmost: original Autoformer with timestamps). For both metrics, it holds that a lower score indicates a better performance, where the best results are **bold**, and the second-best are underlined.

|  | Att bottleneck | | FF bottleneck | | No bottleneck | | *With timestamps* | |
|---|---|---|---|---|---|---|---|---|
|  | MSE | MAE | MSE | MAE | MSE | MAE | MSE | MAE |
| Electricity | 0.275 | 0.374 | 0.695 | 0.589 | **0.215** | **0.325** | 0.280 | 0.368 |
| Traffic | 1.581 | 0.860 | 1.040 | 0.616 | 0.635 | 0.403 | **0.619** | **0.387** |
| Weather | **0.227** | **0.293** | 0.246 | 0.310 | 0.246 | 0.321 | 0.269 | 0.344 |
| Illness | 4.810 | 1.603 | 4.767 | 1.557 | 4.473 | 1.530 | **3.405** | **1.295** |
| Exchange rate | 0.203 | 0.329 | 0.196 | 0.330 | 0.181 | 0.308 | **0.152** | **0.283** |
| ETT | 0.279 | 0.353 | 0.284 | 0.373 | 0.169 | 0.280 | **0.155** | **0.265** |

## N.2    CKA

After training the Autoformer with our bottleneck framework, we evaluate the similarity of its hidden representations to the interpretable concepts using CKA, see Figure 33. Recall that CKA scores range from 0 to 1, where 1 indicates perfect similarity. Both components in the two types of bottleneck show a very high similarity to their concept. Interestingly, the similarity of the hour-of-day component is very high, even though the model did not obtain the timestamps as input. Therefore, we can conclude that the bottleneck components are capable of learning to represent *novel* concepts from the data.

## N.3    Component Visualizations

We visualize the contributions of each component in the bottleneck using the Decoder Lens method (Langedijk et al., 2023), see Figure 34. We obtain the output from each component individually by masking the other components with zero. Each component seems to provide similar contributions to the forecast as their respective counterpart in the previously studied Transformer architectures. In particular, the first component (see Figure 34a) produces forecasts of correct seasonality and few irregularities, similar to the AR model. The second component (see Figure 34b) follows the hour-of-day feature, and the free head (34c) picks up on high-frequency data patterns.

## N.4    Conclusion

In this experiment, we trained an Autoformer without providing timestamps as input, while including a bottleneck component to be similar to the hour-of-day feature. The results from our interpretability analyses

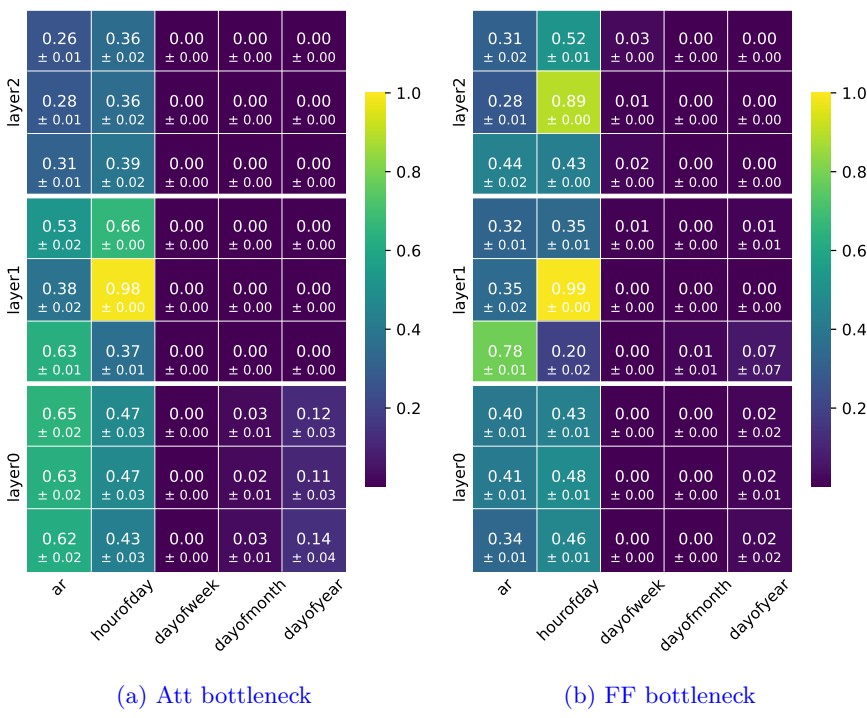

(a) Att bottleneck                    (b) FF bottleneck

Figure 33: CKA scores of the Autoformer's encoder (containing three heads per layer) from the attention and feed-forward bottleneck on the electricity dataset, where each score denotes the similarity of an individual component. The first component of `layer1` is trained to be similar to AR, and the second component to the hour-of-day concept (lower and middle row in the figure, respectively). The scores are calculated using three batches of size 32 from the test data set.

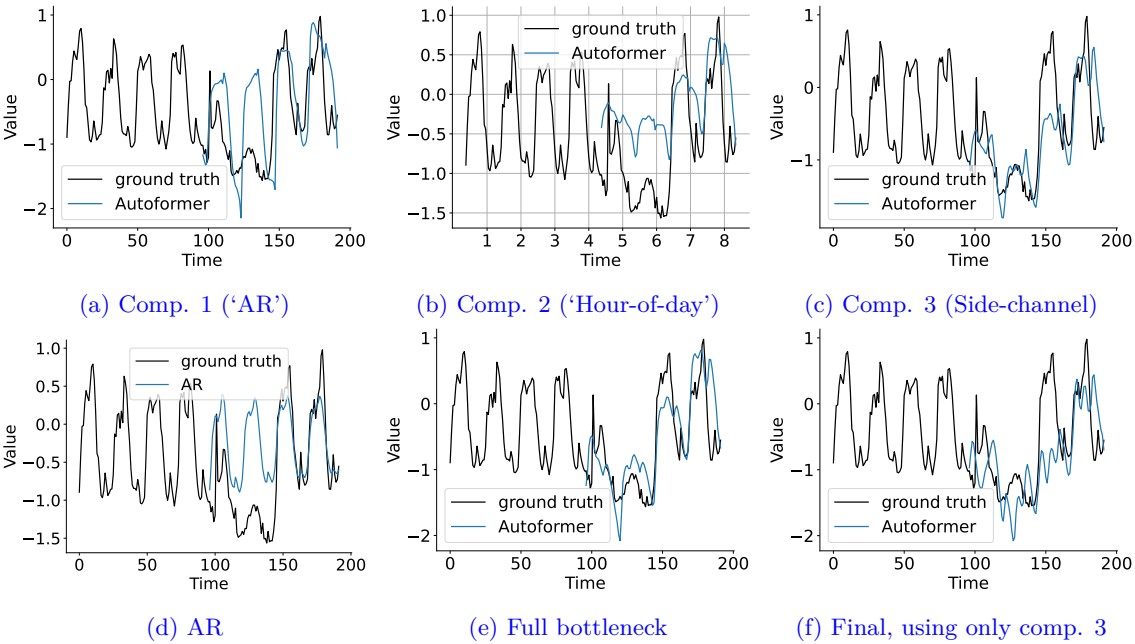

(a) Comp. 1 ('AR')  (b) Comp. 2 ('Hour-of-day')  (c) Comp. 3 (Side-channel)

(d) AR  (e) Full bottleneck  (f) Final, using only comp. 3

Figure 34: Autoformer forecasts from the components in the bottleneck layer (FF bottleneck on electricity data) in 24a, 24b and 24c. They are obtained by masking the other components with zero (the mean). The first half of the ground truth forms the input to the model. Note that the horizontal axes are the same across all figures, but Figure 24b contains a grid of days instead of numbered hours. Figure 24d shows the forecast made by the surrogate model AR; Figure 24e shows the forecast of the entire layer (i.e., all components together), and 24f shows the forecast of the final layer when only the third component is used in the bottleneck layer. Note the difference between Figures 24c and 24f, where we decode from the bottleneck and the final layer, respectively.

show that the bottleneck is able to learn this feature. Therefore, the bottleneck layer can learn to be similar to *novel* concepts, instead of only copying it from the input. Note that we cannot perform the previously introduced intervention experiments, where we shift the timestamps given as input to the model, as these models do not receive timestamps as input.

