# OpenReview forum: "Interpretability for Time Series Transformers using A Concept Bottleneck Framework"
_TMLR — Rejected by TMLR_

### Review · Reviewer_wB21 · 2025-05-16

**Summary Of Contributions:**

The authors propose a framework to enhance the interpretability of Transformer based time series forecasting models. They achieve this by means of a Concept Bottleneck approach, in which the predictor is trained with an additional loss term, based on Centered Kernel Alignment, which pushes the model towards providing interpretable features as intermediate representation in its layers. In particular they propose the usage of a surrogate timestamp and the predictions of a simple Auto Regressive model as the interpretable concepts. They show that the approach results in the hidden representations being aligned to the corresponding concepts and that the performance is comparable to that of an equivalent model without such interpretability mechanism. Moreover they provide an example of intervention to recover performance in case of inputs that are time-shifted w.r.t. the training distribution's timestamps.

**Audience:**

Yes

**Broader Impact Concerns:**

I don't have any broader impact concerns

**Claims And Evidence:**

No

**Requested Changes:**

I suggest that the authors address the following requests:
* Assessing the behavior of the proposed approach when the transformer model does not have **T** as one of its inputs. This would strenghten the interpretability claims, as the model needs to learn the concept from the time series patterns rather than having the option to copy it from the input. Moreover repeating the intervention experiments on a model trained in such manner would ensure that the behavior outlined in the weaknesses cannot take place.
* Expanding the experimental analysis to include more transformer architectures for time series forecasting, like Informer, PatchTST, Reformer or others.
* Assessing the efficacy of the proposed concepts in tasks where the AR model does not excel. If such datasets are difficult to find, a synthetic approach with data generated by a system with non-linear dynamics could still provide valuable insights.
* Improving the readability and clarity of the paper in general. Some examples:
	* Clarifying the notation can help readability. In the current state some of the notation seems pseudo-code, while superscripts and subscripts of different symbols are not clearly introduced. Moreover equations lack numbering.
	* Narrowing down the explanation of how the approach modifies a transformer block to the necessary parts. It appears that the relevant information is that an encoder block consists of a certain number of attention heads followed by a fully connected layer, stripped of any skip-connection, and that you apply concepts alignment either at the output of the attention or at that of the fully connected. Other details seem architecture specific and can be moved to the appendix.
* Clarifying, in Appendix H, how the vanilla transformer is modified and which are its inputs.

Additionally there are some sentences which are unclear, in particular:
* Sec. 2.1: "concepts are represented as a supervised vector". What is a supervised vector?
* Sec. 2.3: what does "embedding to the same space" mean? Is this two linear layers applied to X and T to have the same number of features as output?
* Sec. 4.4: "the definition of the output of any encoder layer remains unchanged and is based on T." Can you clarify this? If T is only provided as input via the embedding layer wouldn't the output of all encoder layers be based on the shifted timestamp?
* Sec. 5: "An interesting direction for future research would be to optimize the number and type of interpretable concepts in the bottleneck, and extend the framework to other modalities." It is unclear what this sentence refers to. What are "other modalities", and what does "optimizing the number and type of concepts" refers to?
* It is not clear what differentiates Tables 3 and 4.
* The caption of Figure 5 is difficult to follow. If having the days on the x-axis in Figure 5b is important can't all figures use that axis? Moreover (this holds for similar figures), drawing a dashed vertical line where the forecasts start can improve the readability.
* Caption of Figure 5 says "They are obtained by masking the other components with zero (the mean)". Maybe this can be made more clear, by stating that you mask whit zeros which are a good enough approximation of the mean.

**Strengths And Weaknesses:**

**Strenghts**
* The proposed methodology is novel in its combination of existing approaches.
* The framework is suitable for transformer based models, which have risen in populairty to solve time series forecasting tasks.
* The suggested bottleneck concepts can be applied to various time series tasks.
* Interpretability of forecasting models is indeed a compelling topic in high stake applications.

**Weaknesses**
* The paper is confusing to read, with some concepts explained in possibly too much detail while others are just touched upon. In general there is not a natural flow in the reading as the level of detail and focus shifts abruptly.
* The idea of a Concept Bottleneck seems that of enforcing a model to pass through interpretable representations to compute its final output. However it appears that in the proposed framework the information flow is not restricted to the bottleneck, as the "free" concept can be used by the model as a bypass. Hence there is no guarantee that the "observability" provided by the interpretable concepts reflects the information used by the model. Moreover, by looking at Table 4, it seems that removing the "free" concept degrades performance.
* It is stated that the proposed approach can turn any time series transformer into a CB transformer, however experiments are limited to two architectures.
* The proposed concepts, while general, are not applicable to every time series task. Think, for instance, of the case of forecasting the behavior of a dynamical system with strong non-linear dynamics. In this case how can the concept of "timestamp" be expressed? Moreover the forecasts of a linear AR model might not be meaningful concepts for such scenario.
* It is unclear to me why the addition of the proposed concepts should make the model more interpretable, in particular when considering that one of the concepts (*hour-of-day*) is provided as input to the model and the other (AR) is a surrogate of the forecasting task itself. While it is easy to see the interpretability of an image classifier which predicts the concepts of "animal", "four-legged" and "household" and then combines them to predict "cat", the interpretability does not seem as straightforward in the time series case. The idea appears clearer in the synthetic dataset, however real-world tasks do not come with a similar decomposition of concepts.
* The authors claim "while the overall performance remains largely unaffected – in many cases surpassing results from the original Autoformer paper", however results from Table 1 seem to suggest otherwise. From my understanding the original Autoformer has a different number of layers, hence the comparison is not completely fair. Conversely, while looking at the performance without any bottleneck, it seems that adding the bottleneck results in performance degradation in most cases, and it is difficult to asses its significancy without the measure of variance across seeds.
* The AR model seems to largely outperform transformers in the majority of datasets. While this is not directly a problem from the perspective of the paper, it raises the question of wether the AR concept would be effective in tasks where a simple AR model does not perform well.
* While I see how the interpretable timestamps can be used to recognize an anomaly in the model behavior, I have some concerns about the applicability of the proposed intervention approach in a real world setting. If we don't know the kind of time-shift that was applied to the input how can we practically intervene on the model?
* The effectiveness of the intervention might be a byproduct of the fact that *hour-of-day* is also provided as an input to the model. This is if the model learns to copy the *hour-of-day* from the input to the concept layer and then "reads" the *hour-of-day* directly from the concept, ignoring the input. In this case the effectiveness of the intervention would be due to the fact that it is almost equivalent to fixing the original input to the model.
* Code for reproducibility is not provided.

**Questions**
- Is there a specific reason to use CKA? Does the choice of similarity metric have a significant impact on the framework?
- Which are the values of *hour-of-day* concept representations when the model is provided with shifted timestamp at the input in the intervention experiment? Do they align with the shifted values of with the original "true" values?
- In Figure 10d (Illness) and 10e (Exchange rate) the forecasts of the AR model appear to significantly deviate from the ground truth, even though its performance in Table 1 is the best one for both datasets. Is it just an unlucky sample or you found similar visual results on other examples from the same two datasets?

---

> ### Author Response · Authors · 2025-06-04
> **Initial reply to questions**
>
> Thank you for your extensive reply. We highly appreciate the time and effort you put into reviewing our work. Upon receiving the full set of reviews, we intend to provide a revised version of the paper containing your requested changes. For now, we would like to clarify your current questions:
>
> 1. *Is there a specific reason to use CKA? Does the choice of similarity metric have a significant impact on the framework?*
>
> The reason for using CKA is that it allows for comparing representations of different dimensionalities. For example, the model dimension of the Autoformer and Vanilla Transformer is 512, while the AR and hour-of-day representations are of smaller dimensionalities. CKA measures the distances in each representation separately, and then compares the similarity structures. Other similarity metrics could be used, but we find that CKA works sufficiently.
>
> 2. *Which are the values of hour-of-day concept representations when the model is provided with shifted timestamp at the input in the intervention experiment? Do they align with the shifted values of the original "true" values*
>
> In the intervention experiment, we patch the concept representations with those of the same model component (in the same layer) on the original, unshifted time data. These activations only align with the shifted activations of the original input if our bottleneck training framework works well. The results show that the original performance can be retrieved almost perfectly in most test cases, and therefore only this model component contains the hour-of-day concept information used by the model.
>
> 3. *In Figure 10d (Illness) and 10e (Exchange rate) the forecasts of the AR model appear to significantly deviate from the ground truth, even though its performance in Table 1 is the best one for both datasets. Is it just an unlucky sample or you found similar visual results on other examples from the same two datasets?*
>
> No, these are not unlucky samples, we find similar visual results on other examples from these datasets. AR predicts the next future value as a linear combination of its past values, which is insufficient for particularly volatile datasets such as Illness and Exchange rate. The Transformer models seem to have learned there is volatility, but still struggle on forecasting, resulting in a higher mean error. In general, forecasts can be evaluated in different ways, and lower mean errors can look visually worse.

---

> ### Author Response · Authors · 2025-06-26
> **Reply to requested changes**
>
> We have now updated the paper, and wrote all new parts in blue. Regarding your requested changes:
> - We have implemented and carried out the experiments you suggested of training the Autoformer model without access to timestamps $\mathbf{T}$ as *input* (see Appendix M; in the CKA-experiment we do encourage *similarity* to the hour-of-the-day feature). As expected, this hurts the overall performance, although the cost is still modest. The hour-of-day component from the bottleneck is still able to achieve high similarity to the hour-of-day feature (CKA score >= 0.98). This shows that the bottleneck layer is able to learn concepts that it cannot simply copy from the input.
> - We did seriously consider the idea of doing an additional intervention experiment in this set-up, but we should point out that simply doing the same intervention is not possible. Our earlier experiment used shifted timestamps as input, but timestamps are no longer part of the input in the new experiments. You mentioned a possible weakness of our paper was that the model may copy the hour-of-day information from the input to the bottleneck layer, to then “read” it directly from the bottleneck layer. In this case, the effectiveness of the intervention would be due to the fact that this is almost equivalent to fixing the original input. However, this is the point of the intervention experiment: to show that the hour-of-day component in the bottleneck is the only model component that is actually *used* by the model for hour-of-the-day information. This point still stands, and we do *not* make any claims that the concept bottleneck is learning novel concepts, instead we claim that our methodology allows us to *localize* the concepts.
> - We have expanded the experimental analysis to include more transformer architectures for time series forecasting, namely FEDformer (see Appendix J).
> - There is an assessment of the efficacy of the proposed concepts in tasks where the AR model does not excel, namely the electricity and traffic dataset. Additionally, we refer to Appendix L on the effect of AR as a surrogate model, where we train the Autoformer without AR as concept in the bottleneck layer.
> - We improved the readability and clarity of the paper. In particular:
>   - Narrowed down the explanation by moving details to the Appendix, which already removes equations with complicated symbols containing superscripts and subscripts.
>   - Changed Figure 2 to be less architecture dependent.
>   - Clarified in Appendix H how the vanilla transformer is used and which are its inputs.
>   - Clarified the unclear sentences.
> - We will publicly share our code if the paper gets published.
>
> We kindly invite you to take another look at the paper, and hope we have convinced you of its value. Please let us know if anything remains unclear. Thank you for your time.

---

> > ### Comment · Reviewer_wB21 · 2025-07-03
> >
> > Dear authors, thanks for the response and for putting effort in addressing my suggestions, however my main concerns still stand. In particular:
> >
> > >* We have implemented and carried out the experiments you suggested of training the Autoformer model without access to timestamps  as *input* (see Appendix M; in the CKA-experiment we do encourage *similarity* to the hour-of-the-day feature). As expected, this hurts the overall performance, although the cost is still modest. The hour-of-day component from the bottleneck is still able to achieve high similarity to the hour-of-day feature (CKA score >= 0.98). This shows that the bottleneck layer is able to learn concepts that it cannot simply copy from the input.
> >
> > While this helps in showing the model does not need to directly copy from the input, Table 7 shows that the model with the bottleneck is performing worse than the one without it in almost all datasets. While this cost could be justifiable in exchange for interpretability, I still fail to see how having some hidden (hence not interpretable) representations aligned with known concepts provides practical utility in this regard.
> >
> > >* We did seriously consider the idea of doing an additional intervention experiment in this set-up, but we should point out that simply doing the same intervention is not possible. Our earlier experiment used shifted timestamps as input, but timestamps are no longer part of the input in the new experiments. You mentioned a possible weakness of our paper was that the model may copy the hour-of-day information from the input to the bottleneck layer, to then "read" it directly from the bottleneck layer. In this case, the effectiveness of the intervention would be due to the fact that this is almost equivalent to fixing the original input. However, this is the point of the intervention experiment: to show that the hour-of-day component in the bottleneck is the only model component that is actually *used* by the model for hour-of-the-day information. This point still stands, and we do *not* make any claims that the concept bottleneck is learning novel concepts, instead we claim that our methodology allows us to *localize* the concepts.
> >
> > I think the fact that intervention cannot be performed in this case is problematic w.r.t. the claimed interpretability of the proposed approach. As the concept representations are not human-interpretable they cannot be directly manipulated to perform intervention. This is a fundamental difference with the original idea behind concept bottlenecks. As the intervention cannot be replicated in a scenario where the model can't "learn to copy", the alternative interpretation remains plausible, as it is an efficient solution to minimize CKA while retaining the input information in a non-redundant way. I think that the fact that the concept is localized in the respective head does not provide much interpretability in practical terms. Moreover, there seems to be no evidence that concept misalignment occurs when the model's predictions are poor. Nonetheless, even if it was the case, one would not have any practical way of intervening to obtain a better prediction.
> >
> > >* There is an assessment of the efficacy of the proposed concepts in tasks where the AR model does not excel, namely the electricity and traffic dataset. Additionally, we refer to Appendix L on the effect of AR as a surrogate model, where we train the Autoformer without AR as concept in the bottleneck layer.
> >
> > The experiment in Appendix L seems counterintuitive to me. If the electricity dataset is the one where the AR model is the worst performer why did you use that to show that removing the corresponding concept does not impact performance (hence it is not the responsible of the improvement)?
> >
> > >* We improved the readability and clarity of the paper.
> >
> > Thanks for making the suggested clarifications. However I would like to point out that simply moving the equations to the Appendix does not solve the readability issues.

---

> > > ### Author Response · Authors · 2025-07-03
> > >
> > > Dear reviewer, thanks for the response. We appreciate your concerns, and we would like to make a few comments:
> > >
> > > > While this helps in showing the model does not need to directly copy from the input, Table 7 shows that the model with the bottleneck is performing worse than the one without it in almost all datasets. While this cost could be justifiable in exchange for interpretability, I still fail to see how having some hidden (hence not interpretable) representations aligned with known concepts provides practical utility in this regard.
> > >
> > > (Mechanistic) interpretability research often provides only partial interpretations of complex models, of which the practical utility can remain unclear. For example: SAEs project the activations of one layer into a (hopefully interpretable) latent vector; circuits identify the most important model components for a specific task; and probes identify whether specific information is present in the activations of one model layer. These methods generally aim to identify *what* the model learns, and *where*, often disregarding the practical utility of this information. The strength of our research is in mechanistically interpreting complex models during training, instead of after (like the popular, post-hoc methods previously mentioned). We kindly disagree with your notion that hidden representations imply non-interpretable representations.
> > >
> > > > I think the fact that intervention cannot be performed in this case is problematic w.r.t. the claimed interpretability of the proposed approach. As the concept representations are not human-interpretable they cannot be directly manipulated to perform intervention. This is a fundamental difference with the original idea behind concept bottlenecks. As the intervention cannot be replicated in a scenario where the model can't "learn to copy", the alternative interpretation remains plausible, as it is an efficient solution to minimize CKA while retaining the input information in a non-redundant way. I think that the fact that the concept is localized in the respective head does not provide much interpretability in practical terms. Moreover, there seems to be no evidence that concept misalignment occurs when the model's predictions are poor. Nonetheless, even if it was the case, one would not have any practical way of intervening to obtain a better prediction.
> > >
> > > - We agree that there is no practical way of intervening when the model’s predictions are poor in general, but that is not the claim of the intervention experiment. The aim is to show that, by construction, the hour-of-day component in the bottleneck is the only model component that is used by the model for this information. This removes the need for post-hoc interpretability of this concept by other means, such as finding an hour-of-day SAE latent, an hour-of-day circuit, or an hour-of-day probe.
> > > - Additionally, as you mention, there *is* a fundamental difference with the original idea of concept bottleneck models. This is why we call our method a framework based on concept bottleneck models. In particular, we do not obtain scores for pre-defined concepts and train a simple model to use these scores for the final task (which we think would be too restrictive for a time-series transformer). This removes some benefits of concept bottleneck models, such as direct manipulation of the concept scores, but we *can* make interesting claims about what the bottleneck learns (and that only the bottleneck is used for this information).
> > >
> > > > The experiment in Appendix L seems counterintuitive to me. If the electricity dataset is the one where the AR model is the worst performer why did you use that to show that removing the corresponding concept does not impact performance (hence it is not the responsible of the improvement)?
> > >
> > > That is a good point. We will perform the experiment on other datasets.
> > >
> > > > Thanks for making the suggested clarifications. However I would like to point out that simply moving the equations to the Appendix does not solve the readability issues.
> > >
> > > Could you be more specific about which readability issues remain? We did not just move equations to the Appendix, but rewrote multiple sections from the paper, and we think that the readability improved.

---

> > > > ### Comment · Reviewer_wB21 · 2025-07-03
> > > >
> > > > >Could you be more specific about which readability issues remain? We did not just move equations to the Appendix, but rewrote multiple sections from the paper, and we think that the readability improved.
> > > >
> > > > My phrasing here was maybe imprecise. I did not intend to say that the authors just moved part of the main text to the appendix. My note was on the fact that the equations, and thus the formalization of the presented methodology are, from what I can see, mostly the same as they were before. In this sense, the readability of the equations has not improved, and it remains difficult to quickly grasp the relevant aspects without ambiguity.
> > > >
> > > > What I was suggesting in my initial review was to simplify the explanation to the most relevant aspects in the main document and possibly move the detailed explanation, for the specific Autoformer architecture, to the Appendix. This as I think that one does not need to know the nuances of the Autoformer to understand your approach. However, in the current iteration, it seems to me that the main document contains a high level description of the approach, while the formal description still remains specific to the Autoformer and it has been moved to the Appendix. On the contrary, I expected a more streamlined set of equations, in the main document, to explain the general idea. Maybe formalizing the methodology around a simplified attention layer and introducing the distinction between applying it at the Attention of FeedForward level only later on.
> > > >
> > > > I hope this clarifies the intent of my comment.

---

> > > > > ### Author Response · Authors · 2025-07-03
> > > > > **Rewrote the equations to be clearer**
> > > > >
> > > > > Thank you for your clarification. We have rewritten Appendix C to contain a concise, yet complete, set of equations to help the reader understand our framework. The main changes are removing the nuances of the Autoformer, and stripping away unnecessary subscripts and superscripts. We think this appendix is a helpful complement to the high-level explanation in the main text. Upon your request, we could move (parts of) it to the main text.

---

### Review · Reviewer_4mMC · 2025-05-27

**Summary Of Contributions:**

The paper proposes a global interpretability method and framework that is based on Concept Bottleneck Models (CBM) for the task of time-series forecasting via Transformers. The authors further perform local intervention to show how the model responds to temporal data shifts in the interpretable concepts. Overall, their models incorporate some pre-defined notions of concepts, making some of their intermediate representations more interpretable, while achieving comparable performance to vanilla models.

**Audience:**

No

**Claims And Evidence:**

No

**Requested Changes:**

- Missing references:
    - “There is a large body of work in the field of explainable AI to make neural networks more interpretable” [...] (page 1)
    - “Vanilla Transformers” [...] (page 1)

- Use the proper citation for CBMs (https://proceedings.mlr.press/v119/koh20a.html).
- Extend evaluation of identified concepts beyond using CKA, e.g., via probing to test task and concept accuracies (see also CBM paper).
- Extend evaluation for the synthetic experiments in Section 4.1 (see previous point).
- Improve the discussion to address limitations of the proposed approach.
- Extend related works to contextualize the work better in the Explainable AI, concept learning and time series forecasting communities.
- Motivation and discussion for why CKA is used are missing, there are many other representational similarity measures, cf. [1].
- The main issue of the paper is the "much improved interpretability" claim, which to me is not sufficiently supported and, given the full complexity of today's Transformer models, not convincing.


[1] Klabunde, M., Schumacher, T., Strohmaier, M., & Lemmerich, F. (2025). Similarity of neural network models: A survey of functional and representational measures. ACM Computing Surveys, 57(9), 1-52.

**Strengths And Weaknesses:**

**Strengths**
- The paper is overall well-written, easy to follow and addresses a relevant and timely challenge.
- The methods and evaluation are clear, covering a sufficiently extensive number of datasets on two models.
- Consideration of both synthetic and real-world data.

**Weaknesses**
- The approach requires pre-defined interpretable concepts, thus limiting the granularity of interpretability, i.e. here to rather broad “hour-of-day” or “day-of-week” concepts. More fine-grained patterns have to be manually included before training, and/or data-driven concepts would have to be pre-engineered.
- The framework and method builds on several existing ideas and thus the novelty of the methodological contribution is limited.
- Mixed overall performance of the CBM approach (as measured via MSE/MAE on Figure 1)
- Interpretability is assessed by computing similarity from intermediate representations, i.e. CKA scores. It has been argued that such internal model states may not be suitable to achieve transparency, see related discussions on attention being limited as explanations [1,2,3 ].
- Lack of evaluation on common explainability metrics, i.e. faithfulness, that allow to evaluate how good the resulting interpretable representations are at explaining the overall model predictions, cf. [4,5].
- Weak discussion that did not address the limitations of the proposed approach.

[1] Sarthak Jain and Byron C. Wallace. 2019. Attention is not Explanation. In Proceedings of the 2019 Conference of the North American Chapter of the Association for Computational Linguistics: Human Language Technologies, Volume 1 (Long and Short Papers), pages 3543–3556, Minneapolis, Minnesota. Association for Computational Linguistics.

[2] Sarah Wiegreffe and Yuval Pinter. 2019. Attention is not not Explanation. In Proceedings of the 2019 Conference on Empirical Methods in Natural Language Processing and the 9th International Joint Conference on Natural Language Processing (EMNLP-IJCNLP), pages 11–20, Hong Kong, China. Association for Computational Linguistics.

[3] Ali, A., Schnake, T., Eberle, O., Montavon, G., Müller, K. R., & Wolf, L. (2022, June). XAI for transformers: Better explanations through conservative propagation. In International conference on machine learning (pp. 435-451). PMLR.

[4] Hedström, A., Weber, L., Krakowczyk, D., Bareeva, D., Motzkus, F., Samek, W., ... & Höhne, M. M. C. (2023). Quantus: An explainable ai toolkit for responsible evaluation of neural network explanations and beyond. Journal of Machine Learning Research, 24(34), 1-11.

[5] Vilone, G., & Longo, L. (2021). Notions of explainability and evaluation approaches for explainable artificial intelligence. Information Fusion, 76, 89-106.

---

> ### Author Response · Authors · 2025-06-04
>
> We thank the reviewer for the praise and constructive criticism. We will, for the revision that we will work on when the reviews are complete, take these comments to heart.
>
> For now, we would just like to point out that we are aware of the discussions about attention as explanation (refs 1-3), and the metrics used to evaluate input attribution techniques (ref 4).
> We indeed focus on internal states, and it’s true that it is very challenging to achieve transparency through the analysis of these internal states, or, in fact, through any other posthoc interpretability effort. But our argument is exactly that (i) if, despite this, Transformer-based models are used (because they yield, for many domains, state-of-the-art results), and (ii) if more transparency is really required, we must try to make the most of a challenging situation. We believe that our focus on human interpretable concepts and the use of CKA in the loss function are real, novel contributions, and have moved us to a different territory, which the arguments from the attention-is-not-explanation debate have little to say about.
>
> Moreover, because of our focus on internal states, the “common explainability metrics” (which are quite problematic anyway) are not applicable, because they have been (mostly) developed for input attribution methods.  However, we do assess the quality of our interpretations in what has quickly become the most common way to assess the quality of interpretations in Mechanistic Interpretability: by doing a causal intervention study (section 4.4), which does provide strong quantitative support that the CKA-analysis from section 4.3.1 did in fact provide a faithful interpretation. In the revision, we will clarify this role of the intervention study in the relation to faithfulness evaluations [1], and hope we can convince the reviewer of its value!
>
> [1] Qing Lyu, Marianna Apidianaki, Chris Callison-Burch; Towards Faithful Model Explanation in NLP: A Survey. Computational Linguistics 2024; 50 (2): 657–723.

---

> ### Author Response · Authors · 2025-06-26
> **Reply to requested changes**
>
> We have now updated the paper and included the requested changes in blue text. We would like to refer back to our previously posted reply. Additionally:
> - We included missing references.
> - We changed the citation for CBMs.
> - Probing experiments on the bottleneck concepts are not trivial, because our bottleneck concepts are higher-dimensional representations (i.e., the hidden representations of AR and the four-dimensional time embeddings). We could probe for each time feature individually (e.g. hour-of-day), but the disadvantage of probes is that it remains unclear whether the model actually uses the probed information. In contrast, causal interventions (causal tracing/activation patching) do make this clear. By performing our intervention experiment on the bottleneck’s hour-of-day component, we verify that this component is the only one that represents the hour-of-day information that the model uses. We included an extra figure in Section 4.4 (Figure 6) to better explain the intervention experiment. This experiment is stronger than probing for hour-of-day information, because we causally test it.
> - We improved the discussion to address limitations.
> - We extended the related works.
> - We included a motivation for the use of CKA.
> - The “much improved interpretability” claim stems from the fact that we can localize where the transformer model stores information, which was not clear earlier. In our bottleneck layer, we generally know what each component represents. We know this, in particular, due to (1) the high CKA scores, (2) the component visualizations and (3), most importantly, the intervention experiment which shows that the bottleneck’s hour-of-day component is the only model component that represents the hour-of-day information that is used by the model.
>
> We hope that we have convinced you of our approach’s value. Please let us know if anything  still remains unclear. Thank you for your time.

---

### Review · Reviewer_KAJY · 2025-06-22

**Summary Of Contributions:**

This work develops an interpretability method for transformers for time-series applications. The paper proposes a new training objective based on Concept Bottlenecks using Central Kernel Alignment. The idea here is to introduce a loss function at an intermediate layer of the transformer with similarity metric with respect to some interpretable concepts. The paper provides experiments on Vanilla Transformer and Autoformer and test of real and synthetic datasets. Apart from providing interpretability, the proposed model also provides competitive performance compared to models that are not interpretable and also allows for intervention.

**Audience:**

Yes

**Claims And Evidence:**

Yes

**Requested Changes:**

Please check the weakness section for the requested changes.

**Strengths And Weaknesses:**

Strengths:
- The paper introduces a novel training method for interpretability in time series transformer using a Concept Bottleneck framework
- The paper provides experiments on Vanilla Transformer as well as Autoformer on real and synthetic datasets showing that even though the model is interpretable, the performance remains competitive to the method without interpretability
- The paper further shows how to use the bottleneck layer for intervention and modifying the model behavior

Weaknesses:
- The experiments are focused on very small transformer models with only three layers. Further, the experiments are focused only on a specific kind of bottleneck concepts. Does the results (interpretability without drop in performance for example) hold even for larger models? Additionally are there other concept bottlenecks that can help interpretability in time series models?
- Section 4.4 is not very clear to me. Is it possible to divide the section 4.4 into further paragraphs and clearly explain the nature of intervention and results separately. I found the definition and notations somewhat confusing. Also, please further explain what other interventions are possible with the current framework and if it works with larger models as well.
- In section 4.4, what is the difference between "no intervention" and "original" label in Fig. 6. The captions can be improved, use consistent naming in figures/tables and in text. E.g., in Sec.4.2 it says "AR surrogate model", but in Table 1, I do not see "AR surrogate model". Can you explain?

---

> ### Author Response · Authors · 2025-06-26
> **Reply to questions**
>
> Thank you for your review! Regarding your requested changes:
>
> - The size of the time series transformers, in the long-term forecasting literature, tend to be small. They are often trained with only *two encoder layers* and *one decoder layer*, so we decided to train our bottleneck framework on transformers of a similar size (*three encoder layers* and *one decoder layer*). We expect the results (i.e., interpretability without a drop in performance) to also hold for larger models, because there are even more layers to ‘recover’ from the bottleneck. Additionally, there can be many other concept bottlenecks that could help interpretability, depending on the desired interpretable concepts. We have opted for domain-agnostic concepts in the bottleneck (AR surrogate model and hour-of-day feature), but one could imagine including domain-specific concepts (e.g. a speech or guitar feature for music time series). Additionally, we have chosen for different concepts in the experiment with synthetic data to show the generality of our bottleneck framework.
>
> - To meet your request to improve the explanation of the intervention experiment, we have included a new figure in Section 4.4 (Figure 6). In short, we run the transformers on time-shifted data, but replace the activations of the hour-of-day component in the bottleneck with those from a run on the original input. Figure 7 shows the results of the intervention, namely that the intervention restores the original performance in most settings. This indicates that only the hour-of-day component in the bottleneck represents the hour-of-day information that is used by the transformer, and therefore the bottleneck is trained successfully.
>
> - As mentioned in the caption of Figure 7, “original” represents the transformer on the original data (without a timeshift), while “no intervention” represents the transformer when run on time-shifted data, without executing the intervention (i.e. not replacing the activations of the hour-of-day component). Additionally, as we explain in Section 3.2, AR (autoregression) is the surrogate model. Table 1 contains the results of AR in the corresponding column.
>
> In the revised paper, we have marked all major changes in response to reviewer comments in blue. Please let us know if anything remains unclear.

---

### Author Response · Authors · 2025-07-15
**Any remaining unclarities?**

Dear reviewers,

If we have read the TMLR instructions correctly, there are only about
5 more days left for discussion, and hence for us to convince you that
the evidence we have provided really supports the claims we make.
The paper is dear to us -- we believe it really breaks new ground, by
showing how we can move, in the domain of long-term time series, from
posthoc interpretability to successful *enforcement* of
interpretability during training.

We hope the interesting detailed discussions on time series
transformers vs. classical models, and on what counts as a successful
interpretation, don't overshadow that bigger point. We'd appreciate
your feedback if there are remaining unclarities or points of
disagreement!

Best wishes, the authors

---

### Decision · Action_Editor_4fed · 2025-08-11

**Recommendation:** Reject

**Additional Comments:**

The topic is important and will be of interest to the TMLR audience.

Although all reviewers agreed that the work is timely and can be of wide interest if developed further, 2 reviewers argued that the work is still not ready for publication at TMLR. Along with the points given above, here are the major points of contention pointed out by the reviewers:

* The model is not restricted to using the inferred concepts to perform the prediction and there is no clear evidence that errors in the concepts' prediction relate to increased prediction error. Thus, advantages of interpretability are not clear in this case.

*  Lack of multiple seed runs makes it hard to asses the likelihood of negative impact on performance.

* The experiments focus on small-scale models and few pre-defined concept settings, which limits the scope of the paper.

* The key motivation of using CKA over others metrics remains unresolved.

I looked at the paper and I agree with the reviewers. I really liked the core idea of the work but am of the opinion that the work requires a major revision at this stage to be ready for publication. I thus encourage the authors to take the reviews into account and submit again. Good luck with the future iterations of the submission.

**Audience:**

Yes

**Audience Explanation:**

The topic is important and will be of interest to the TMLR audience.

**Claims And Evidence:**

No

**Claims Explanation:**

The reviewers agreed that there is not enough evidence currently for the work to warrant acceptance. For example, the concepts remain hidden behind a latent representation and intervention cannot be performed in the practical case stripping away the advantage of concept bottleneck models. Also, the framework aims at localizing pre-defined and known concepts, thereby making the claim of overall interpretability a weak one.

**Resubmission Of Major Revision:**

The authors may consider submitting a major revision at a later time.